# UI2Code$^N$: UI-to-Code Generation as Interactive Visual Optimization

**Zhen Yang** [*1]   **Wenyi Hong** [*1]   **Mingde Xu** [2]   **Xinyue Fan** [2]   **Weihan Wang** [2]   **Jiale Cheng** [1]   **Xiaotao Gu** [2]   **Jie Tang** [1]

## Abstract

UI-to-code aims to translate UI screenshots into executable front-end code. Despite progress with vision-language models (VLMs), most existing methods formulate UI-to-code as a single-pass generation, which mismatches real-world UI development that is inherently iterative and feedback-driven. We reformulate UI-to-code as an **interactive visual optimization** problem, where code generation is embedded in a closed-loop process of execution, visual inspection, and iterative refinement driven by rendered visual feedback. To address the non-differentiability of visual objectives and the noise of absolute visual evaluators, we propose **Relative Visual Policy Optimization (RVPO)**, a preference-based reinforcement learning method that optimizes relative visual rankings among rendered candidates under execution feedback. We instantiate this paradigm in **UI2Code$^N$**, an open-source 9B model trained via continual pre-training, supervised fine-tuning, and reinforcement learning. Experiments demonstrate state-of-the-art performance on UI drafting, UI polishing, and UI editing benchmarks, even outperforming larger models, with performance consistently improving through iterative visual optimization. Our code and models are available at https://github.com/zai-org/UI2Code_N.

## 1. Introduction

Recent advances in visual language models (VLMs) have substantially expanded the range of problems that can be addressed directly from visual inputs. Among them, the task of translating UI screenshots into executable front-end code, i.e. UI-to-code, has become practically feasible. As user interfaces constitute a central component of modern software systems, reliable UI-to-code automation promises significant impact, from reducing software engineering development cost to broadening access to application creation.

As illustrated in Figure 1, real-world UI development is inherently iterative and feedback-driven. Rather than producing code in a single pass, developers draft an initial implementation, render it, visually inspect discrepancies, and refine the code based on executable feedback. This closed-loop process continues until the rendered UI sufficiently matches the target design.

However, most existing UI-to-code approaches predominantly model the task as a single-turn generation problem, aiming to predict executable code in one pass from a visual specification. This formulation fundamentally mismatches real-world UI development workflows. This mismatch arises from three fundamental properties of UI coding. First, rendered UI behavior cannot be reliably inferred from code alone, as runtime factors such as browser defaults, font fallback, and DPI scaling introduce non-trivial deviations. Second, visual discrepancies are multi-factorial and tightly coupled (e.g., spacing, alignment, and typography), making one-shot prediction inherently brittle. Third, rendered feedback provides an explicit and observable self-verification signal that enables test-time correction and iterative improvement. These properties jointly suggest that successful UI-to-code systems must explicitly reason over executable feedback and support iterative policy improvement, rather than rely on one-shot token prediction.

Motivated by these observations, we reconceptualize UI-to-code as an **interactive visual optimization** problem. More broadly, we consider a class of learning problems where models generate executable artifacts, while the optimization objective is defined in a non-differentiable execution space and can only be accessed through rendered or simulated outcomes. UI-to-code represents a concrete instantiation of this setting, where the objective is visual fidelity under rendering. Rather than treating UI-to-code as a purely token-space generation task, we explicitly optimize for visual correctness in the rendering space and use executable feedback as the supervision signal. Under this formulation, UI drafting, UI

---
[*]Equal contribution [1]Department of Computer Science and Technology, Tsinghua University [2]Zhipu AI. Correspondence to: Jie Tang <jietang@tsinghua.edu.cn>.

*Proceedings of the 43$^{rd}$ International Conference on Machine Learning*, Seoul, South Korea. PMLR 306, 2026. Copyright 2026 by the author(s).

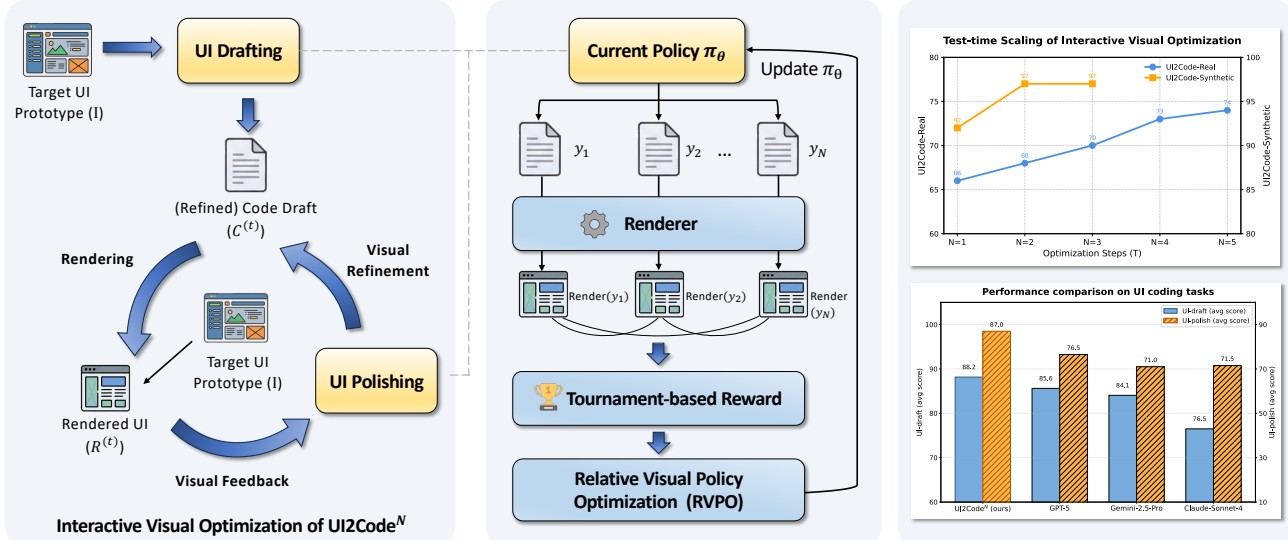

*Figure 1.* **Left:** Interactive visual optimization in UI2Code[N]. The VLM first performs UI drafting to generate an initial code draft $C^{(0)}$, which is rendered into $R^{(0)}$. Using visual feedback from the rendering, the same VLM iteratively performs UI polishing to produce refined code $C^{(t)}$. **Middle:** Relative Visual Policy Optimization (RVPO), the proposed reinforcement learning algorithm used to optimize both UI drafting and UI polishing. **Right (top):** Performance consistently improves with additional refinement steps, highlighting the iterative nature of real-world UI development. **Right (bottom):** Quantitative results on UI-to-code generation and UI polishing benchmarks.

polishing (visual refinement), and UI editing are unified as different instantiations of the same optimization loop.

This perspective also exposes a key technical challenge: the optimization objective is evaluated in a rendered visual space and is therefore non-differentiable with respect to output tokens. Although one may assign an absolute visual score to each rendered candidate using a VLM-based judge, such scores are often poorly calibrated and exhibit high variance across samples, leading to unstable learning signals when used directly as rewards. To address this challenge, we propose **Relative Visual Policy Optimization (RVPO)**, a preference-based reinforcement learning approach that derives rewards from relative visual comparisons among multiple rendered candidates. By optimizing relative rankings rather than absolute scores, RVPO yields a more stable learning signal and naturally aligns with the comparative nature of visual evaluation.

We instantiate this paradigm in UI2Code[N], an open-source 9B vision–language model trained via (i) continual pre-training, (ii) supervised fine-tuning, and (iii) reinforcement learning with RVPO. Despite having only 9B parameters, UI2Code[N] achieves state-of-the-art performance across UI-to-code generation, UI polishing, and UI editing benchmarks, outperforming both open- and closed-source models with substantially larger parameter counts. Moreover, iterative refinement consistently improves visual fidelity, empirically validating the optimization-driven nature of executable UI development. While UI2Code[N] represents a concrete instantiation, the proposed interactive visual op-

timization paradigm and relative optimization strategy are model-agnostic and broadly applicable to executable generation problems with black-box feedback.

In summary, our contributions are three-fold:

- We formulate UI-to-code as an instance of **interactive visual optimization**, a learning paradigm for executable generation under non-differentiable, execution-based feedback.

- We propose **Relative Visual Policy Optimization (RVPO)**, a reinforcement learning objective that optimizes implicit visual correctness via group-wise relative preference rather than unstable absolute rewards.

- We instantiate this paradigm in UI2Code[N], a compact 9B open-source model that achieves state-of-the-art performance on UI-to-code generation, UI polishing, and UI editing benchmarks, outperforming significantly larger closed-source systems.

**Conflict of Interest Disclosure.** Some authors are affiliated with Zhipu AI, which develops foundation models related to the GLM family. This paper evaluates GLM-based models and uses GLM-4.5V as part of the visual reward verification pipeline. These affiliations and model usages are disclosed for transparency. All comparisons are conducted following the protocols described in the paper.

# 2. Method

## 2.1. UI-to-Code as Interactive Visual Optimization

We formulate UI-to-code as an interactive visual optimization problem, where the objective is defined over rendered UI behavior under executable feedback. Unlike one-shot generation, UI code quality can only be reliably assessed after execution, as runtime factors (e.g., layout engines, font fallback, DPI scaling) introduce discrepancies not observable from static code.

From this perspective, UI-to-code is best viewed as an **interactive visual optimization problem**, where code is iteratively refined based on executable rendering feedback. As illustrated in Figure 2, this process forms a closed-loop feedback system rather than a single-pass mapping from images to code.

We formalize this process as a feedback-driven transformation:

$$\mathcal{F}_\theta(I, C, R, E) \to C', \tag{1}$$

where $I$ denotes the target UI image, $C$ the current code, $R = \text{Render}(C)$ the rendered output, $E$ optional modification instructions, and $C'$ the updated code. Existing UI-to-code methods correspond to a degenerate case of this formulation, where the transformation is applied only once and rendering feedback is ignored.

Under this formulation, drafting, refinement, and editing are not distinct tasks, but different instantiations of the same optimization process, characterized by how feedback and constraints are introduced.

**Definition of Visual Optimization.** For clarity, we define *visual optimization* as the process of iteratively improving executable code such that its rendered visual output increasingly aligns with a target UI. Formally, given a target image $I$, the objective is to find code $C^\star$ that minimizes an implicit visual discrepancy:

$$C^\star = \arg\min_C \ \mathcal{D}(I, \text{Render}(C)), \tag{2}$$

where $\text{Render}(\cdot)$ denotes a black-box, non-differentiable rendering process, and $\mathcal{D}$ represents an implicit visual distance that can only be accessed through execution and visual comparison. Importantly, visual optimization in this work does not assume differentiability or explicit gradients. Instead, it refers to an optimization-inspired, feedback-driven process that iteratively proposes, evaluates, and refines code based on rendered visual feedback.

## 2.2. Instantiations of Visual Optimization

**UI Drafting.** The optimization process is initialized by UI drafting, which produces a first-pass code approximation

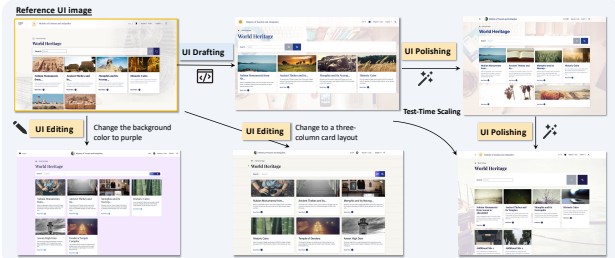

*Figure 2.* UI-to-code as an interactive visual optimization process. Code is iteratively refined based on executable rendering feedback.

from the target UI:

$$C^{(0)} = \mathcal{F}_\theta(I). \tag{3}$$

Drafting captures the global layout and major visual structures but cannot resolve discrepancies arising from rendering-dependent behavior. As such, it serves as a cold start rather than a complete solution.

**UI Polishing (Visual Refinement).** Given a draft $C^{(t)}$, UI polishing iteratively improves code quality by explicitly comparing rendered output against the target UI:

$$C^{(t+1)} = \mathcal{F}_\theta(I, C^{(t)}, R^{(t)}), \quad R^{(t)} = \text{Render}(C^{(t)}). \tag{4}$$

This refinement loop performs feedback-driven policy improvement, progressively reducing visual discrepancies such as misalignment, spacing errors, or style inconsistencies. While the objective is non-differentiable and only observable through rendering, repeated refinement enables **test-time scaling**, where larger iteration budgets lead to higher visual fidelity. We denote this scalable process as UI2Code$^N$.

**UI Editing.** Editing is treated as a conditional variant of refinement, where code updates are guided by explicit modification instructions:

$$C' = \mathcal{F}_\theta(I, C, E). \tag{5}$$

In this work, UI editing focuses on localized, well-specified UI changes and does not aim to solve general instruction-driven UI design.

## 2.3. Relative Visual Policy Optimization

Under the visual optimization formulation, the learning objective is defined over rendered UI outcomes, which are non-differentiable with respect to the generated code. For real-world complex UIs, ground-truth code is frequently unavailable, making likelihood-based training insufficient for aligning generation with visual correctness. We therefore treat UI generation as black-box policy optimization under visual feedback and optimize the policy with reinforcement learning.

A natural approach is to train a reward model that assigns an absolute reconstruction score to each rendered UI. In practice, however, absolute scores produced by model-based evaluators (e.g., VLM judgers) are often noisy, poorly calibrated, and sensitive to prompt or scale variations, which makes absolute reward modeling unstable for policy optimization. To address this challenge, we propose **Relative Visual Policy Optimization (RVPO)**, which derives rewards from relative visual preference among multiple rendered candidates, eliminating the reliance on calibrated absolute scores.

**Relative Preference Surrogate Objective.** Let $\pi_\theta(\cdot \mid x)$ denote the code-generation policy under input context $x$. Given two candidates $(y, y')$, we assume access to a visual judger $\mathcal{C}_\psi$ that provides a pairwise preference. Further details of the visual judger, including prompting and evaluation protocols, are provided in Appendix A. We model the judger as inducing a (possibly stochastic) preference probability

$$p_\psi(y \succ y' \mid x) := \mathbb{P}[\mathcal{C}_\psi(x, y, y') = 1], \qquad (6)$$

where the randomness may arise from the judger itself or ambiguity in visual comparison. Instead of optimizing absolute visual scores, RVPO is guided by the following rank-based surrogate objective:

$$\mathcal{L}_{\text{rank}}(\theta) = \mathbb{E}_{y \sim \pi_\theta(\cdot \mid x)} \left[ \mathbb{E}_{y' \sim \pi_\theta(\cdot \mid x)} \left[ p_\psi(y \succ y' \mid x) \right] \right], \quad (7)$$

which measures the expected preference of a policy sample against policy-induced alternatives. This objective directly reflects the selection goal in UI generation, where the desired output is the visually best candidate among a set of alternatives.

**Tournament-based Reward Construction.** The inner expectation in Eq. (7) is intractable in practice. We therefore approximate it using group-wise sampling and tournament aggregation. Specifically, we sample $N$ candidates $\{y_i\}_{i=1}^N \sim \pi_\theta(\cdot \mid x)$ and perform pairwise comparisons within the group. For each ordered pair $(i, j)$, $i \neq j$, we define the binary outcome

$$o_{ij} := \mathbf{1}[\mathcal{C}_\psi(x, y_i, y_j) = 1]. \qquad (8)$$

Each candidate $y_i$ is then assigned a scalar reward based on its aggregate win count,

$$W_i := \sum_{j \neq i} o_{ij}, \qquad (9)$$

which summarizes its relative preference within the sampled group. This tournament-based aggregation reduces sensitivity to noise in individual comparisons and avoids

dependence on absolute score calibration. In practice, we apply simple monotonic transformations of $W_i$ (e.g., normalization and fixed penalties for failed renders) as the final scalar reward, as detailed in Algorithm 1.

---

**Algorithm 1** Visual Relative Reward at Iteration $t$

---

**Input:** visual optimization iteration $t$, target UI $I_{\text{target}}$, rollout renders $\{I_{t,i}\}_{i=1}^N$; optional $I_{t-1,\text{ref}}$ (UI-polishing)
**Output:** $\text{Reward}[t, i]$
$\text{Pool} \leftarrow \emptyset$
**for** $i = 1$ **to** $N$ **do**
  **if** $I_{t,i}$ fails to render **then**
    $\text{Reward}[t, i] \leftarrow -1$
    **continue**
  **end if**
  **if** UI-polishing **then**
    $(S_{\text{ref}}, S_i) \leftarrow \text{comp\_score}(I_{\text{target}}, I_{t-1,\text{ref}}, I_{t,i})$
    **if** $S_i < S_{\text{ref}}$ **then**
      $\text{Reward}[t, i] \leftarrow 0$
      **continue**
    **end if**
  **end if**
  $\text{Pool} \leftarrow \text{Pool} \cup \{i\}$
**end for**
**for each** $i \in \text{Pool}$ **do**
  Compute $(S_i, S_j) = \text{comp\_score}(I_{\text{target}}, I_{t,i}, I_{t,j})$ **for all** $j \in \text{Pool}, j \neq i$
  $\text{Reward}[t, i] \leftarrow$
  $1 + \sum_{j \in \text{Pool}, j \neq i} \left( \mathbf{1}[S_i > S_j] + \frac{1}{2}\mathbf{1}[S_i = S_j] \right)$
**end for**

---

We denote by $(t, i)$ the $i$-th rollout at visual optimization iteration $t$. Here, $t = 0$ corresponds to UI drafting that produces the initial code, while $t \geq 1$ denotes successive UI polishing iterations. Tournament-based ranking is our default reward design; simpler alternatives are evaluated in ablation studies in Section 4.3.

**Efficiency of Pairwise Comparison.** Although RVPO relies on pairwise visual comparisons, this step is not the computational bottleneck in practice. Candidate rollouts are generated and rendered once, while VLM-based comparisons are performed only as post-hoc scoring over successfully rendered outputs. These comparisons are independent and can be parallelized across workers, and tournament aggregation further restricts comparisons to candidates that pass the rendering check. Empirically, pairwise comparison accounts for only 2.2% of the wall-clock time per RVPO training iteration, with most of the cost coming from autoregressive generation and rendering. A detailed wall-clock breakdown is provided in Appendix G.

**Policy Optimization with GRPO.** RVPO applies Group Relative Policy Optimization (GRPO) (Shao et al., 2024) to perform stable policy improvement under tournament-based rewards. Given rewards $\{r_i\}_{i=1}^N$ within a group, GRPO

computes group-normalized advantages

$$A_i = \frac{r_i - \bar{r}}{\sigma_r}, \qquad (10)$$

and updates the policy using the clipped surrogate objective

$$\mathcal{J}(\theta) = \mathbb{E}\left[ \frac{1}{N} \sum_{i=1}^{N} \min(\rho_i A_i, \ \mathrm{clip}(\rho_i, 1 - \epsilon, 1 + \epsilon)A_i) \right], \qquad (11)$$

where $\rho_i = \pi_\theta(y_i \mid x)/\pi_{\theta_{\mathrm{old}}}(y_i \mid x)$. We omit KL regularization to avoid overly constraining policy updates when optimizing implicit visual objectives.

## 3. Training Pipeline

Formulating UI-to-code as visual optimization imposes requirements beyond standard vision–language pretraining. The model must perceive fine-grained UI elements, reason over long structured code sequences, and iteratively improve rendered behavior under implicit visual objectives.

To meet these requirements, we adopt a three-stage training pipeline that progressively aligns perception, reasoning, and optimization: (i) continual pre-training for vision–code grounding, (ii) supervised fine-tuning for learning refinement behavior, and (iii) reinforcement learning for direct visual optimization. This section outlines the training pipeline and optimization objectives, with detailed data construction and implementation settings provided in Appendix B. All training data are collected from publicly accessible sources or existing datasets with permissive licenses. The standard ethical guidelines are provided in Appendix C.

### 3.1. Continual Pre-training

Continual pre-training establishes foundational alignment between UI images and their corresponding DOM structures. We optimize an autoregressive next-token prediction objective over code conditioned on UI images, providing perceptual grounding for downstream visual optimization.

To strengthen localized grounding, we adopt a GUI-REG–style objective (Hong et al., 2024). Given a UI image $I$ and a bounding box $b$ corresponding to a DOM node, the model predicts the associated code snippet $C_b$ via:

$$\mathcal{L}_{\mathrm{dom}}(\theta) = \mathbb{E}_{(I,C),\,b} \left[ -\sum_{n=1}^{|C_b|} \log p_\theta(c_{b,n} \mid c_{b,<n}, I, b) \right]. \qquad (12)$$

To preserve global coherence, we additionally optimize a standard image–code likelihood objective:

$$\mathcal{L}_{\mathrm{pair}}(\theta) = \mathbb{E}_{(I,C)} \left[ -\sum_{n=1}^{|C|} \log p_\theta(c_n \mid c_{<n}, I) \right]. \qquad (13)$$

### 3.2. Supervised Fine-tuning

Supervised fine-tuning instantiates the visual optimization paradigm across UI drafting, UI polishing, and instruction-conditioned editing. To encourage explicit reasoning over visual discrepancies, model outputs are structured as:

$$\texttt{<think>}\, T \,\texttt{</think><answer>}\, C' \,\texttt{</answer>}, \qquad (14)$$

where $T$ captures intermediate diagnosis and $C'$ contains executable code.

We optimize the likelihood of the thought-augmented sequence:

$$\mathcal{L}_{\mathrm{SFT}}(\theta) = \mathbb{E}_{(\mathcal{X}, T, C')} \left[ -\log p_\theta(T, C' \mid \mathcal{X}) \right], \qquad (15)$$

with inputs $\mathcal{X}$ varying across drafting, refinement, and editing settings.

### 3.3. Reinforcement Learning

Reinforcement learning enables direct optimization for visual alignment beyond token-level likelihood. We apply Relative Visual Policy Optimization (RVPO) (Section 2.3) after supervised fine-tuning, ensuring stable refinement behavior and mitigating reward hacking. All reinforcement learning data and reward signals are disjoint from evaluation benchmarks.

## 4. Experiments

### 4.1. Evaluation Setup

**Benchmarks.** We evaluate UI2Code[N] on several established UI-to-code benchmarks, including Design2Code (Si et al., 2024), Flame-React-Eval (Ge et al., 2025), and Web2Code (Yun et al., 2024). As these benchmarks mainly consist of relatively simple screenshots, we additionally construct *UI2Code-Real*, a benchmark of 115 real-world webpages collected from in-the-wild sources, to assess generalization under realistic UI complexity. For the UI polishing task, we introduce *UIPolish-bench*, which contains 100 synthetic and 100 real-world webpages, enabling evaluation across both controlled and in-the-wild settings. Further details of all benchmarks are provided in Appendix D.

**Evaluation Metrics and Reliability.** We consider two evaluation paradigms: (1) **CLIP-based scoring**, which measures semantic visual similarity following prior work (Si et al., 2024), and (2) **VLM-based scoring**, which leverages visual large language models to provide human-aligned judgments of design fidelity and usability (Yun et al., 2024). Unless otherwise specified, we primarily report VLM-based evaluation results, as our empirical analysis shows that VLM-based rewards better align with human preferences

*Table 1.* Experimental results on UI-to-Code and UI Polishing benchmarks. **Bold** text indicates the best score among open-source models, and underlined text indicates the best score across all models.

| Model | UI Drafting | | | | UI Polishing | |
|---|---|---|---|---|---|---|
| | Design2Code | Flame | Web2Code | UI2Code-Real | UIPolish-Real | UIPolish-Synthetic |
| **Open-source VLM** | | | | | | |
| InternVL3-9B | 15.3 | 11.3 | 12.3 | 16.5 | 4.0 | 7.0 |
| InternVL3-78B | 30.0 | 51.3 | 45.5 | 30.4 | 10.0 | 15.0 |
| Qwen3-VL-8B-Thinking | 56.6 | 56.3 | 68.7 | 49.7 | 32.1 | 41.0 |
| Qwen3-VL-32B-Instruct | 83.3 | 82.5 | 81.2 | 65.0 | 46.0 | 55.0 |
| Qwen2.5-VL-7B | 29.1 | 25.0 | 37.2 | 26.1 | 11.0 | 14.0 |
| Qwen2.5-VL-72B | 41.9 | 46.3 | 64.1 | 40.9 | 23.0 | 38.0 |
| MiMo-VL-7B-SFT | 28.3 | 10.0 | 44.3 | 33.9 | 17.0 | 33.0 |
| MiMo-VL-7B-RL | 28.7 | 8.8 | 38.3 | 30.4 | 16.0 | 30.0 |
| Kimi-VL-A3B-Instruct | 27.3 | 50.0 | 69.1 | 26.1 | 14.0 | 40.0 |
| Kimi-VL-A3B-Thinking | 38.8 | 36.3 | 46.6 | 27.0 | 14.0 | 27.0 |
| GLM-4.1V-9B-Thinking | 64.7 | 72.5 | 71.3 | 53.0 | 42.0 | 46.0 |
| **Closed-source VLM** | | | | | | |
| Claude-4-5-Sonnet-Thinking | 82.9 | 92.5 | 87.8 | 67.2 | 81.0 | 66.0 |
| Claude-4-Sonnet-Thinking | 81.2 | 76.3 | 85.1 | 63.5 | 78.0 | 65.0 |
| Claude-3.7-Sonnet-Thinking | 77.7 | 80.0 | 73.3 | 55.8 | 75.0 | 62.0 |
| GPT-5 | 89.7 | 91.3 | 93.7 | 67.8 | 85.0 | 68.0 |
| GPT-4o | 35.3 | 75.0 | 62.7 | 21.7 | 26.0 | 14.0 |
| o4-mini | 63.8 | 83.8 | 77.9 | 59.1 | 65.0 | 65.0 |
| Gemini-2.5-Pro | 89.5 | 87.5 | 90.6 | 68.7 | 74.0 | 68.0 |
| Gemini-2.5-Flash | 70.5 | 72.5 | 85.7 | 62.6 | 17.0 | 24.0 |
| Doubao-1.5-Thinking-Vision | 53.7 | 78.8 | 55.6 | 38.3 | 51.0 | 61.0 |
| Doubao-1.6-Thinking-250715 | 62.4 | 67.7 | 67.2 | 43.4 | 61.0 | 67.0 |
| UI2Code$^N$-9B-SFT | 79.3 | 85.0 | 80.8 | 67.0 | 76.0 | 89.0 |
| UI2Code$^N$-9B-RL | **88.6** | **95.0** | **92.5** | **76.5** | **80.0** | **94.0** |

and yield more stable optimization than CLIP-based similarity (Section 4.3). The reliability of VLM-based evaluation is further validated through human agreement studies and variance analysis (Section 4.6). For UI polishing, we adopt a pairwise evaluation protocol. Given an initial rendering $B$ and a refined rendering $C$ with respect to a target UI $A$, polishing is considered successful if $C$ is preferred over $B$. We report polishing accuracy as the fraction of instances where refinement leads to a clear improvement. More details of evaluation metrics are provided in Appendix E.

## 4.2. Main Results

**Results with VLM Scoring.** We first evaluate UI2Code$^N$ using VLM-based scoring, which provides human-aligned judgments of visual fidelity, layout correctness, and overall UI quality under executable rendering. Table 1 reports results on both UI drafting and UI polishing benchmarks, including public datasets and our curated benchmarks. Across all UI drafting tasks, UI2Code$^N$ consistently outperforms open-source VLMs and remains competitive with leading closed-source systems, with particularly strong gains on UI2Code-Real, which contains long and structurally complex webpages. On UI polishing, existing open-source VLMs fail to achieve reliable performance, while UI2Code$^N$-9B-RL attains 94.0% accuracy on UIPolish-Synthetic and 80.0% on UIPolish-Real, matching or exceeding closed-source models. These results support our hypothesis that UI coding benefits from an interactive

optimization paradigm rather than single-pass generation.

**Results with CLIP Scoring.** We additionally report CLIP-based evaluation results on the Design2Code benchmark. Table 2 summarizes CLIP similarity scores together with component-level metrics measuring block structure, text content, spatial position, and color consistency. Overall, strong vision-language models achieve comparable CLIP scores, indicating accurate reconstruction of coarse visual appearance. However, CLIP shows limited sensitivity to structural and layout-level discrepancies. Models with substantially different block and position accuracy often obtain similar CLIP scores, suggesting that CLIP primarily captures global visual similarity rather than fine-grained structural correctness. Consistently, UI2Code$^N$ improves block- and position-level metrics without a corresponding increase in CLIP score. This observation indicates that CLIP-based evaluation alone is insufficient for assessing structural and functional correctness in UI coding tasks, motivating the use of VLM-based evaluators.

**Interactive Visual Optimization with UI Polishing.** A key implication of formulating UI coding as an interactive visual optimization problem is the ability to perform test-time scaling via iterative refinement. UI2Code$^N$ first generates an initial implementation and then progressively improves it using rendered execution feedback. As shown in Table 3, UI-to-code performance consistently improves with more refinement rounds $N$ on both real and synthetic

*Table 2.* Comparison of CLIP-based visual and component-level structural metrics on the Design2Code benchmark.

| Model | Block | Text | Position | Color | CLIP |
|---|---|---|---|---|---|
| GPT-5 | 89.1 | 94.2 | 86.4 | 78.0 | 81.6 |
| Gemini-2.5-Pro | 89.1 | 93.5 | 85.5 | 71.4 | 80.9 |
| Claude-4-Sonnet | 88.7 | 93.2 | 84.6 | 72.0 | 80.5 |
| Qwen2.5-VL-72B | 86.6 | 91.6 | 76.8 | 67.8 | 77.8 |
| UI2Code$^N$-SFT | 86.8 | 91.5 | 81.7 | 69.7 | 79.0 |
| UI2Code$^N$-RL | 88.7 | 93.1 | 83.8 | 72.6 | 80.5 |

benchmarks. While performance on UI2Code-Synthetic saturates early ($N = 3$), UI2Code-Real continues to benefit from additional refinement up to $N = 5$, reflecting the higher structural complexity of real-world webpages. These results validate that executable feedback enables effective test-time scaling under the interactive visual optimization formulation.

*Table 3.* Test-time scaling performance of interactive UI-to-code generation

| Benchmark | N = 1 | N = 2 | N = 3 | N = 4 | N = 5 |
|---|---|---|---|---|---|
| UI2Code-Real | 66.0 | 68.0 | 70.0 | 73.0 | 74.0 |
| UI2Code-Synthetic | 92.0 | 97.0 | 97.0 | – | – |

**Comparison with Agent-based Systems.** We further compare UI2Code$^N$ with representative agent-based UI-to-code systems on Design2Code-HARD. Unlike DCGen and ScreenCoder, which decompose UI-to-code into multiple detection, planning, and generation stages, UI2Code$^N$ performs end-to-end generation and refinement under executable visual feedback. As shown in Table 4, UI2Code$^N$-RL achieves higher VLM-judge accuracy than both agent-based baselines, while requiring substantially lower latency and token cost. These results suggest that interactive visual optimization provides a simpler and more efficient alternative to heavily engineered multi-agent pipelines.

*Table 4.* Comparison with agent-based UI-to-code systems on Design2Code-HARD.

| Model | VLM Acc. | Latency (s) | Token Cost |
|---|---|---|---|
| DCGen | 45.0 | ∼137 | ∼7600 |
| ScreenCoder | 80.0 | ∼66 | ∼4600 |
| UI2Code$^N$-RL | **88.6** | **∼40** | **∼2600** |

### 4.3. Ablation Study: The Impact of Reward Design

We study the effect of reward design on reinforcement learning for both UI polishing and UI drafting. All ablation experiments start from the UI2Code$^N$-SFT checkpoint and use identical RL configurations (batch size 32, rollout size 16, learning rate 1e−6), isolating the impact of reward formulation.

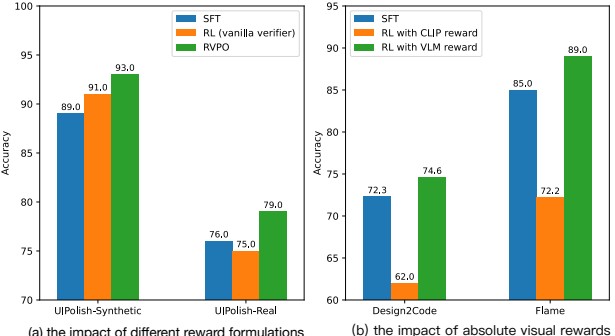

*Figure 3.* The impact of reward design.

**Reward Design for UI Polishing.** To justify the effectiveness of RVPO, we conduct an ablation study on reward design. Specifically, we compare: (i) an absolute (vanilla) reward that assigns independent scores to each candidate without relative comparison, and (ii) the full RVPO reward based on tournament-style aggregation, as described in Section 2.3. As shown in Figure 3(a), RVPO consistently outperforms both supervised fine-tuning (SFT) and simpler reinforcement learning baseline with a vanilla verifier on the UI polishing task. While RL with an absolute verifier yields limited improvement over SFT on the UIPolish-Synthetic benchmark, it fails to consistently outperform SFT on the UIPolish-Real benchmark. In contrast, RVPO achieves the highest accuracy on both UIPolish-Synthetic and UIPolish-Real. These results indicate that optimizing relative visual preference among multiple candidates is more effective for UI polishing than relying on absolute reward scores.

**Reward Design for UI Drafting.** We further analyze reward design for UI drafting under reinforcement learning with a vanilla verifier by comparing two absolute visual similarity signals: CLIP-based similarity and VLM-based visual judgments. As shown in Figure 3(b), the choice of visual similarity signal leads to substantially different outcomes. Using CLIP similarity as the reward consistently degrades performance relative to SFT on both Design2Code and Flame, indicating that global semantic alignment is insufficient for guiding fine-grained UI drafting. In contrast, VLM-based absolute rewards provide moderate improvements over SFT, but still underperform relative to RVPO. These results underscore the sensitivity of UI drafting to reward design and further motivate relative, preference-based optimization for robust learning.

**Reward Hacking Analysis.** A potential concern is that RVPO may exploit visual feedback by producing visually aligned but structurally brittle code, such as overusing hard-coded absolute positioning. To verify this, we measure the ratio of absolute-positioned elements in generated HTML/CSS on Design2Code. As shown in Table 5, this ratio remains low after reinforcement learning and

slightly decreases from 0.7% for UI2Code$^N$-SFT to 0.5% for UI2Code$^N$-RL. This suggests that RVPO does not obtain its gains through trivial absolute-positioning shortcuts, but improves rendered UI quality while preserving structurally reasonable code.

*Table 5.* Absolute positioning ratio on the Design2Code benchmark.

| Model | Absolute Positioning Ratio |
|---|---|
| UI2Code$^N$-SFT | 0.7% |
| UI2Code$^N$-RL | 0.5% |

### 4.4. Ablation Study: The Impact of Real-world Webpages in RL Stage

We further study the effect of incorporating real-world webpages during the reinforcement learning (RL) stage of UI2Code$^N$. We conduct a controlled comparison using identical RL data budgets (20k samples) and training steps (100 iterations), differing only in whether real-world webpages are included. While synthetic webpages offer clean supervision and controlled UI patterns, they may not fully capture the visual complexity, noise, and distributional diversity of real-world interfaces. To account for this gap, we augment the RL stage with a curated set of in-the-wild webpages, where target UI screenshots are collected from real-world sources. As shown in Figure 4, incorporating real-world webpages during RL consistently improves performance across both UI drafting and UI polishing benchmarks. The gains are particularly pronounced on evaluations involving real-world webpages, indicating improved alignment with realistic UI distributions. These results suggest that real-world data plays a critical role during visual policy optimization, complementing synthetic data and enabling more effective sim-to-real transfer in UI coding tasks.

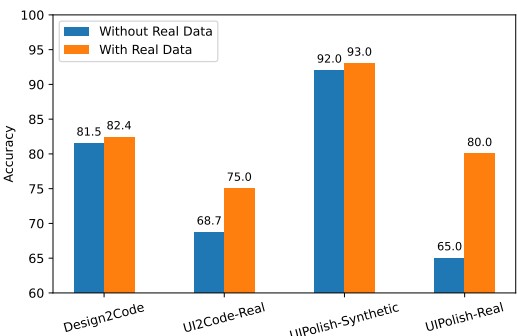

*Figure 4.* The impact of real-world webpages in RL stage.

### 4.5. Ablation Study: The Impact of Training Stages

We further analyze the contribution of each training stage in UI2Code$^N$ on the Design2Code benchmark. As shown in Table 6, continual pre-training (CPT) improves accuracy from 9.0% to 53.3%, indicating that large-scale UI

*Table 6.* Impact of different training stages on UI-to-code performance on the Design2Code benchmark.

| Training Stage | Accuracy (%) | Gain |
|---|---|---|
| Base Model | 9.0 | – |
| + Continual Pre-training | 53.3 | +44.3 |
| + Supervised Fine-tuning (SFT) | 79.3 | +26.0 |
| + Reinforcement Learning (RL) | 88.6 | +9.3 |

image–code pairs provide essential vision–code grounding. Supervised fine-tuning (SFT) further improves performance to 79.3% by aligning the model with UI-to-code instructions and structured output formats. Finally, reinforcement learning with RVPO improves accuracy to 88.6%, showing that executable visual feedback provides complementary gains beyond supervised learning. These results suggest that the three-stage pipeline progressively equips the model with UI perception, instruction following, and feedback-driven visual optimization.

### 4.6. Human Evaluation and Judge Validation

To validate the reliability of VLM-based evaluation and reward signals, we conduct comprehensive human studies and judge consistency analyses on UI-to-code evaluation.

**Human Evaluation on Design2Code-HARD.** Figure 5 shows human preference results on the Design2Code-HARD benchmark. Across all comparisons, UI2Code$^N$ consistently outperforms strong open-source models and remains competitive with leading closed-source systems, including GPT-5, Gemini-2.5-Pro, and Claude-4-Sonnet. Detailed win/tie/lose statistics and VLM-based judge results are reported in Appendix F.1.

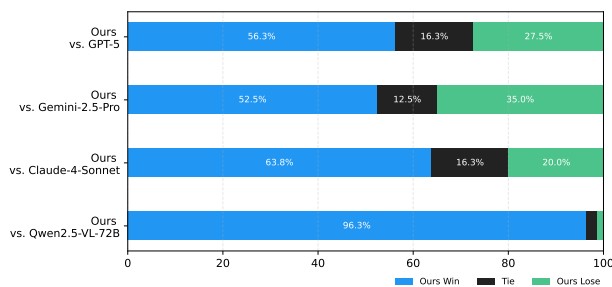

*Figure 5.* Human evaluation on Design2Code-HARD.

**Human Evaluation on UI Editing.** We conduct human evaluation on UIEdit-Bench, a benchmark of 69 UI editing tasks covering diverse layouts and editing instructions. Human annotators rate edited UIs on a 0–5 scale in terms of edit correctness, preservation of unedited regions, and overall quality. As shown in Table 7, the RL-enhanced UI2Code$^N$ model achieves the highest scores across all criteria, outperforming both commercial and open-source baselines. These

results indicate that reinforcement learning with relative visual feedback substantially improves instruction-following accuracy and structural preservation in UI editing.

*Table 7.* Human evaluation results on UIEdit-Bench (0–5 scale).

| Model | Edit Correctness | Preservation | Overall |
|---|---|---|---|
| Claude-4-Sonnet | 4.83 | 4.54 | 4.69 |
| Gemini-2.5-Pro | 4.42 | 4.17 | 4.30 |
| GPT-5 | 4.63 | 4.46 | 4.54 |
| Qwen-2.5-VL-72B | 3.53 | 3.30 | 3.41 |
| UI2Code[N]-SFT | 4.64 | 4.57 | 4.60 |
| UI2Code[N]-RL | **4.94** | **4.80** | **4.87** |

**Human–VLM Alignment Analysis.** We validate the reliability and human alignment of our evaluator via two complementary studies on Design2Code-HARD (80 samples), covering both model-level decision consistency and sample-level score calibration. At the decision level, VLM-based pairwise comparisons closely match human preferences across all baselines, achieving strong correlation with human judgments (Pearson 0.93, Spearman 1.0) and reproducing the exact same baseline ranking. Notably, the evaluator behaves conservatively relative to humans, indicating no bias toward our model. At the score level, VLM scores correlate with human ratings (Pearson 0.65) and effectively separate good and bad outputs with large effect sizes. More detailed protocols and statistics (e.g., sample-score level and model-decision level) are provided in Appendix F.2. The analysis of evaluator variance and stability is reported in Appendix F.3.

### 4.7. Additional Experiments

Beyond the main results and ablation studies, Appendix G provides additional analyses of evaluator robustness, held-out metric cross-verification, the effect of evaluator choice, the impact of the `Think/Answer` output format, comparisons with specialized UI-to-code models, and oscillations in the UI polishing process. These analyses further examine the stability of our evaluation protocol and the behavior of iterative visual refinement. Appendix H presents qualitative and quantitative case studies illustrating how UI2Code[N] improves generated UIs through multi-round interactive refinement.

### 5. Related Work

**UI-to-Code Benchmarks.** Design2Code (Si et al., 2024) introduced the first large-scale UI-to-code benchmark built from real-world webpages, together with visual-centric metrics such as Block-Match and CLIP similarity. Its construction pipeline simplifies raw HTML by removing external dependencies and replacing images with placeholders, which preserves real sources but yields webpages simpler than those encountered in practice. Subsequent bench-

marks, including Web2Code (Yun et al., 2024) and Flame-React (Ge et al., 2025), refined data pipelines but continued to rely heavily on LLM-synthesized HTML. More recently, WebGen-Bench (Lu et al., 2025) expanded evaluation to functional website generation by employing automated agents to test interactivity and execution behavior.

**UI-to-Code Datasets.** Progress in UI-to-code generation has been largely driven by dataset scaling. Early efforts relied primarily on synthetic data, such as Web-Sight (Laurençon et al., 2024), which generated millions of screenshot–code pairs using Tailwind CSS, and Web2Code (Yun et al., 2024), which combined LLM-synthesized data with curated resources for instruction tuning. Later datasets, including WebCode2M (Gui et al., 2025) and Vision2UI (Gui et al., 2024), sourced data from real-world webpages (e.g., Common Crawl) and applied extensive pruning and filtering. While these datasets preserve basic structure, aggressive pruning often removes dependencies such as CSS, resulting in oversimplified webpages that deviate from realistic UI distributions.

**UI-to-Code Models and Systems.** Although large vision–language models (VLMs) perform well on many multi-modal tasks, they often struggle with UI-to-code generation, producing incomplete or non-compilable code. Early standalone models, such as Pix2Code (Beltramelli, 2018), ScreenAI (Baechler et al., 2024), SightSeer (Laurençon et al., 2024), Flame (Ge et al., 2025), and WebCode2M (Gui et al., 2025), were trained on synthetic data and showed limited generalization. More recent work leverages commercial VLMs through agent-based pipelines, including DECLARUI (Zhou et al., 2024), DCGen (Wan et al., 2025), and ScreenCoder (Jiang et al., 2025), which decompose UI-to-code into detection, planning, and generation stages. While effective in some cases, these systems incur high complexity and latency and remain sensitive to error propagation across modules.

### 6. Conclusion

We presented UI2Code[N], a vision–language model that formulates UI-to-code as an *interactive visual optimization* problem under executable feedback. To optimize the non-differentiable visual objective, we proposed Relative Visual Policy Optimization (RVPO), which constructs stable reward signals from relative visual preferences among rendered candidates. We instantiate this paradigm in an open-source 9B model trained via continual pre-training, supervised fine-tuning, and reinforcement learning with RVPO. Extensive experiments show that UI2Code[N] achieves state-of-the-art performance on UI-to-code generation, UI polishing, and UI editing, outperforming substantially larger open- and closed-source models.

## Acknowledgments

This work was supported by the Natural Science Foundation of China (62425601), Fundamental and Interdisciplinary Disciplines Breakthrough Plan of the Ministry of Education of China (No. JYB2025XDXM101), the National Natural Science Foundation of China (62506195), China Postdoctoral Science Foundation (2025M771572), China Postdoctoral Program for Innovative Talents (BX20250381), the new cornerstone Science Foundation through the XPLORER PRIZE and a research fund from Daimler Greater China Ltd. and Tsinghua University Joint Institute for Sustainable Mobility.

## Impact Statement

UI2Code$^N$ studies automated UI code generation. By formulating UI-to-code as an interactive visual optimization problem, our approach leverages executable visual feedback to improve the reliability and visual fidelity of generated UI code. We expect UI2Code$^N$ to benefit software engineering practice by reducing manual front-end development effort and lowering the barrier to application creation, particularly for visually driven interfaces.

To support the development and evaluation of foundational UI coding capabilities, this work involves the use of publicly accessible webpage resources, including screenshots paired with corresponding HTML/CSS code. All resources are derived from URL seeds in the publicly available Common Crawl index, and their collection and use strictly adhere to applicable policies, including compliance with `robots.txt`, protection of personally identifiable information, and respect for copyright and licensing constraints. Additional details are provided in Appendix C.

At the same time, automated UI generation may be misused to clone existing websites or reproduce protected visual designs. We therefore release the model for research purposes and encourage users to respect website ownership, licensing terms, and applicable copyright restrictions.

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

# A. Reward Verifier Implementation

In this section, we present implementation details of the visual reward verifier used by Relative Visual Policy Optimization (RVPO) in Section 2.3. The verifier serves as the visual comparator $\mathcal{C}_\psi$ for evaluating rendered UI outputs under executable feedback. We describe the evaluator model, prompting strategy, and validation of evaluator reliability.

## A.1. Reward Verifier for UI Drafting

To provide a reward signal for UI drafting in reinforcement learning stage, we employ GLM-4.5V as the visual reward verifier. We choose GLM-4.5V for its strong multimodal reasoning capability and open-source availability, which enables local deployment, stable evaluation, and reproducibility. Given a target UI screenshot $I_{\text{target}}$ and a rendered UI image $I_{\text{cand}}$ produced from generated code, the verifier outputs a similarity score in the range $[0, 100]$, reflecting overall visual fidelity under execution. The reward for UI drafting is normalized to $[0, 1]$ and used as a scalar reward. This continuous formulation provides fine-grained feedback compared to binary success signals, leading to more stable policy optimization.

$$r = \frac{\text{score}(I_{\text{target}}, I_{\text{cand}})}{100}, \quad r \in [0, 1]. \qquad (16)$$

The prompt used for visual verification is shown in *Prompt of Reward Verifier for UI Drafting*.

## A.2. Comparator-Based Verification via Supervised Fine-Tuning

While absolute visual scoring provides useful feedback, RVPO relies on relative visual preference between candidate renderings. We therefore implement the visual comparator $\mathcal{C}_\psi$ using a triplet-based evaluation scheme, where the verifier compares two rendered UIs relative to the same target. We observe that zero-shot VLMs exhibit limited reliability for fine-grained relative comparison. To address this, we apply supervised fine-tuning (SFT) to GLM-4.5V to improve its robustness and calibration for comparative judgments.

**Data Curation:** We construct a dataset of 10k UI-code triplets consisting of a reference UI and two generated renderings sampled from diverse UI-to-code models. We use Gemini-2.5-Pro as a teacher model to annotate relative preferences based on visual alignment and structural correctness. Ambiguous samples with low teacher confidence are filtered, resulting in a high-quality dataset of 8,500 triplets.

**Training Procedure:** The verifier is fine-tuned using a learning rate of $2 \times 10^{-5}$, a batch size of 64, and a packed sequence length of 32k for 200 training iterations. After fine-tuning, the verifier demonstrates substantially improved relative judgment accuracy.

> **Prompt of Reward Verifier for UI Drafting**
>
> You will be given two images:
>
> The first image is the reference image (design draft or target rendering).
>
> The second image is the code rendering, which is generated based on the first image using HTML/CSS/frontend code.
>
> Your task is as follows:
>
> Compare the overall similarity between the two images, on a scale from 0 to 100:
>
> - 0 means completely dissimilar.
>
> - 100 means perfectly identical.
>
> When scoring, you should comprehensively consider the following aspects:
>
> - Layout (whether the structural positions are consistent)
>
> - Color scheme (whether the colors are faithfully reproduced)
>
> - Typography (font, font size, line spacing, etc.)
>
> - Spacing and alignment (whether element spacing and alignment are accurate)
>
> - Fine details (button styles, icons, shadows, borders, etc.)
>
> Strictly follow the output format below:
>
> First, provide the final score, where the value **must** be enclosed in LaTeX `\\boxed{}`. Then, provide a justification for the score, explaining which aspects are similar, which aspects differ, and the main factors influencing the score.

**Validation of Verifier Reliability:** To validate the reliability of the fine-tuned visual verifier, we conduct human–model agreement studies on a held-out validation set of 100 samples. Human annotators independently assess relative visual quality for UI drafting and UI polishing outputs.

As shown in Table 8, our comparator-based verifier achieves 94% agreement with human judgments, compared to 62% for the zero-shot verifier. For reference, a strong commercial model (GPT-5) achieves 85% agreement under the same evaluation protocol. These results demonstrate that supervised fine-tuning substantially improves the accuracy and robustness of the visual comparator used in RVPO.

*Table 8.* Agreement between VLM-based judgments and human evaluations.

| Model | Agreement with Human |
|---|---|
| Base GLM-4.5V (Zero-shot) | 62% |
| Commercial SOTA (GPT-5) | 85% |
| Our comparator-based verifier | 94% |

### A.3. Reward Verifier for UI Polishing

For UI polishing, we adopt a triplet-based visual verification protocol that evaluates whether a refined rendering improves upon an initial draft with respect to a target UI. Given the reference screenshot $A$, the initial rendering $B$, and the polished rendering $C$, the visual verifier compares both $B$ and $C$ against $A$.

The verifier produces similarity scores $\text{score}(A, B)$ and $\text{score}(A, C)$ in the range $[0, 100]$, which are used internally to determine relative improvement. Specifically, a polishing step is considered successful if

$$\text{score}(A, C) > \text{score}(A, B). \tag{17}$$

The reward for UI polishing is defined as a binary signal:

$$r = \begin{cases} 1, & \text{if } \text{score}(A, C) > \text{score}(A, B), \\ 0, & \text{otherwise.} \end{cases} \tag{18}$$

This formulation directly reflects the objective of UI polishing, namely whether the refinement leads to a clear visual improvement over the initial rendering, rather than the absolute reconstruction quality. This binary, relative reward formulation yields a stable and task-aligned learning signal for UI polishing under executable feedback.

The prompt used for UI polishing verification is shown in *Prompt of Reward Verifier for UI Polishing*.

## B. Additional Training Details

In this section, we provide implementation details for the three-stage training pipeline described in Section 3, including data composition, optimization settings, and reinforcement learning configurations.

### B.1. Continual Pre-training Details

Continual pre-training is initialized from an early checkpoint of GLM-4.1V-9B-Base. The training corpus consists of approximately 10M webpage image–HTML pairs collected via large-scale crawling, with URL seeds drawn from the publicly available Common Crawl dataset (Common Crawl Foundation, 2007–). To preserve global rendering fidelity, we additionally incorporate high-quality UI–code datasets, including WebCode2M (Gui et al., 2025) and related curated sources (Laurençon et al., 2024).

---

**Prompt of Reward Verifier for UI Polishing**

You will be given three images:

- The first image is the reference design (target screenshot).

- The second and third images are code renderings generated based on the reference.

Your task is as follows:

1. Assign a similarity score (0–100) to both the second and third images with respect to the reference:

- 0 = completely dissimilar.

- 100 = perfectly identical.

When scoring, consider the following dimensions with approximate weights:

- Layout structure (30%): element positions, alignment, and overall layout.

- Color fidelity (25%): background, text, button colors, etc.

- Typography (20%): font size, weight, spacing, line height, etc.

- Spacing ratios (15%): margins, paddings, and spacing between elements.

- Element details (10%): button corners, borders, icon styles, etc.

- Ignore differences in actual image content (e.g., photos, icons), and only evaluate style fidelity.

2. Provide a brief justification for each score:

- List 2–3 major differences and explain why they affect the score.

- If the rendering is highly consistent, state the reasons (e.g., "layout and colors are almost identical").

3. Provide a final conclusion: indicate which rendering (second or third) is closer to the reference.

- The conclusion **must** be enclosed in LaTeX \\boxed{}.

---

Training optimizes a mixture of the localized DOM grounding objective (Eq. 12) and the global image–code likelihood objective (Eq. 13). Coding data is interleaved with general vision–language tasks to retain broad VLM capabilities. We train with a learning rate of $2 \times 10^{-5}$, a tensor parallel size of 2, and a global batch size of 1,536. The continual

pre-training stage covers approximately 20M vision–code samples in total.

## B.2. Supervised Fine-tuning Details

The supervised fine-tuning (SFT) dataset contains approximately 80K high-quality samples spanning UI drafting, UI polishing, and instruction-conditioned editing. To ensure data fidelity, we adopt a reverse-engineering strategy: we first generate complex ground-truth HTML and then synthetically derive corresponding refinement inputs or editing instructions. This process ensures that the reasoning trace implies a valid causal path to the target code correction.

SFT is performed for 5 epochs with a maximum sequence length of 32,768 tokens. We use a packed batch size of 256 and a learning rate of $5 \times 10^{-6}$. All SFT experiments share the same output structure with explicit reasoning blocks, as described in Section 3.

## B.3. Reinforcement Learning Details

Reinforcement learning is applied only after the completion of supervised fine-tuning to ensure stable drafting and refinement behavior. We adopt Relative Visual Policy Optimization (RVPO) (Section 2.3) to optimize for executable visual alignment.

The reinforcement learning dataset consists of approximately 12K real-world examples and 30K synthetic examples. Real-world data is sourced from Mind2Web, while synthetic samples are generated through a combination of programmatic transformations and iterative visual optimization. To enhance robustness and reduce sensitivity to specific prompting or judging styles, input prompts are diversified using multiple VLMs, including GLM-4.5V and Claude-3.5-Sonnet, as well as iterative refinement with UI2Code$^N$, where $N \sim \mathcal{U}[1,4]$.

We train with a batch size of 64 and a group size of $G = 16$ for 400 optimization steps. All reinforcement learning data and reward signals are strictly disjoint from evaluation benchmarks to avoid contamination.

## C. Ethics Statement

This work investigates UI-to-code modeling and involves constructing a large corpus of webpage screenshots and corresponding HTML/CSS code solely for training purposes. We summarize our data governance, privacy protection, and licensing considerations below.

**Data Collection.** URL seeds from the publicly available Common Crawl index are used only to locate webpages. The webpage content used for training is collected independently by our crawler, which strictly respects

`robots.txt`, domain-level crawling policies, and rate limits. We do not access login-protected, paywalled, or user-specific content.

**Privacy and PII Protection.** To protect user privacy, we apply automated and rule-based filtering procedures to remove pages containing personally identifiable information (PII), such as names, emails, phone numbers, session-dependent content, or user account information. Pages that include sensitive or identifiable user data are discarded before training.

**Copyright and Licensing.** Webpages may contain copyrighted material owned by their respective authors. To mitigate copyright-related risks, we exclude domains whose Terms of Service disallow automated crawling or derivative processing, avoid collecting or redistributing images or other protected assets, and do not release any raw webpage content (including screenshots or source code). Only the trained model is released, which captures general structural and stylistic patterns rather than verbatim webpage content. Our model is trained on top of an Apache-2.0 licensed base model and is released under a research-only, non-commercial license.

**Data Decontamination and Leakage Prevention.** We take several steps to reduce potential overlap between training data and evaluation benchmarks. During data construction, we perform image-level deduplication with hash-based filtering to remove near-duplicate webpage screenshots. This reduces the chance that visually similar samples appear across training and evaluation splits. In addition, all reinforcement learning data and reward signals are kept disjoint from the evaluation benchmarks. The benchmarks are constructed independently from the training corpus, and benchmark samples are not used during continual pre-training, supervised fine-tuning, or reinforcement learning. These procedures are intended to mitigate potential data leakage and ensure that the reported results reflect generalization rather than memorization of benchmark instances.

**Intended Use.** The model is released exclusively for research purposes to support reproducibility and future development in UI-to-code modeling. It is not intended for applications involving sensitive personal data, high-stakes decision making, or commercial deployment without further licensing review.

We believe these measures align with community standards for responsible data governance and ethical AI development.

# D. Details of Benchmarks

Here we illustrate the details of benchmarks that we evaluate on, along with our curated *UIPolish-bench* and *UI2Code-Real*. To ensure a fair comparison between open-source and closed-source systems on our proposed benchmarks, we evaluate a diverse set of models. Specifically, we select five groups representative open-source VLMs, such as InternVL3 (Zhu et al., 2025), Qwen2.5-VL (Bai et al., 2025), MiMo-VL (Team et al., 2025a), Kimi-VL (Team et al., 2025b), and GLM-4.1V-9B-Thinking (Hong et al., 2025). For closed-source systems, we evaluate 4 widely-used models: Claude-4 (Anthropic, 2025), Gemini-2.5 (Comanici et al., 2025), Doubao (Guo et al., 2025), and GPT-5 (OpenAI, 2025). This setup allows us to benchmark UI-to-code and UI polishing performance across both research and industrial systems under the same evaluation protocol.

## D.1. Existing Benchmarks

- Web2Code (Yun et al., 2024): this benchmark comprises 1,198 webpage screenshot images to evaluate the ability of HTML code generation for a multi model. Different from traditional code-level evaluations, this benchmark assesses the generated webpage's fidelity at the image level. This evaluation method converts the predicted HTML codes back into images using Selenium WebDriver to allow a direct visual comparison with the ground truth images.

- Flame-React-Eval (Ge et al., 2025): a benchmark of 80 curated design-to-React cases. In the original evaluation, the generated code is judged correct if it compiles, renders without error, and the rendered screenshot matches the reference with a DINOv2 embedding cosine similarity above threshold.

- Design2Code (Si et al., 2024): contains 484 real-world webpages (plus an 80-example HARD subset) as input screenshots. Models must output corresponding HTML/CSS. The original evaluation is done via rendered visual similarity (CLIP) plus element-level matching (position, text, color), with human judgments used to validate metrics.

## D.2. Our Proposed Benchmarks

Almost all the existing benchmarks are constructed with synthetic or heavily pruned HTMLs, and none of them can evaluate the UI polishing ability. To analyze the UI-to-code and UI-polish capability on real-world webpage distribution, we propose the following benchmarks.

- UI2Code-Real: A benchmark consisting of 115 real-world webpage screenshots. Unlike synthetic datasets, which typically feature simplified layouts and over-pruned structures, UI2Code-Real directly reflects the complexity, visual diversity, and noise inherent in real webpages. This benchmark therefore provides a more realistic and challenging setting for evaluating UI-to-code generation models.

- UIPolish-bench: A benchmark specifically designed to evaluate UI polishing. Each sample consists of a reference screenshot $A$, an initial rendering $B$, and the corresponding HTML/CSS code used to produce $B$. The goal of UI polishing is to compare $A$ and $B$, identify the discrepancies between them, and modify the underlying HTML/CSS code so that the rendered result better aligns with $A$. This design directly captures the iterative refinement process of UI development. UIPolish-bench is further divided into two subsets: 1) **UIPolish-Synthetic**: constructed from synthetic webpages with controlled structures, which ensures clean annotations and facilitates fine-grained evaluation of polishing behavior. 2) **UIPolish-Real**: collected from real-world webpages, which preserves noise, complex layouts, and design diversity, providing a challenging benchmark for assessing polishing in practical settings.

# E. Evaluation Metrics

In this section, we provide more details of the evaluation metrics and prompting protocols used in our experiments.

## E.1. CLIP-based Metrics

CLIP-based evaluation follows prior work (Si et al., 2024) and measures semantic visual similarity between rendered webpages. Given a generated webpage and its reference rendering, we compute CLIP similarity on the full-page image. In addition to the global CLIP score, we report several component-level metrics to capture different aspects of visual fidelity, including block structure, text content, spatial position, and color consistency.

## E.2. VLM-based Evaluation

VLM-based evaluation employs a visual language model as an automated judge to assess UI quality under executable rendering. The judge observes the rendered webpage and provides a scalar score reflecting overall visual fidelity and layout correctness

For the UI drafting task, we employ `o4-mini` as the visual evaluator to assess the fidelity of generated renderings. Given the reference screenshot $A$ and the rendering $B$ generated from the predicted HTML/CSS code, `o4-mini` outputs a similarity score $score(A, B)$ in the range $[0, 100]$, where higher values indicate greater visual resemblance.

To obtain a robust evaluation metric, we define the final accuracy as the proportion of samples whose similarity score exceeds a threshold of 80:

$$\text{Accuracy} = \frac{1}{N} \sum_{i=1}^{N} \mathbb{1}\{\text{score}(A_i, B_i) \geq 80\}, \quad (19)$$

where $N$ denotes the total number of evaluated UI examples. This threshold-based criterion ensures that only renderings with sufficiently high fidelity to the reference are considered successful.

For the UI polishing task, we employ `Gemini-2.5-Pro` as the visual evaluator. The model is prompted with a triplet comparison: a reference screenshot $A$, an initial rendering $B$, and a polished rendering $C$. It is asked to assign similarity scores in the range $[0, 100]$ to both $B$ and $C$, provide brief reasoning for each score, and determine which rendering is closer to the reference.

### E.3. Judge Prompt Templates

All visual evaluators are prompted with standardized instructions to ensure consistency across tasks. Below we provide the prompt templates used for UI drafting and UI polishing tasks.

## F. Evaluator Reliability and Stability

In this section, we report detailed human evaluation results and VLM-based judge comparisons on the Design2Code-HARD benchmark, which consists of 80 challenging UI samples curated to highlight performance differences among state-of-the-art UI-to-code systems.

---

**Prompt for UI Drafting**

You will be given two images:
- The first image is the reference screenshot (design draft or target rendering).
- The second image is the rendering generated from the first image using HTML/CSS/frontend code.
Your task is to evaluate the similarity between the two images and assign a score on a scale from 0 to 100:
- 0 means completely dissimilar.
- 100 means perfectly identical.
The output must follow the required format:
1. Provide the final score, where the value **must** be enclosed in LaTeX `\\boxed{}`.
2. Provide a short justification, explaining the key similarities and differences that influenced your score.

---

**Prompt for UI Polishing**

You will be given three images:
- The first image is the reference (target design draft).
- The second and third images are code-rendered results based on the reference.
Please complete the following tasks:
1. Assign a score to both the second and third images, with a range of 0–100:
- 0 means completely dissimilar to the reference.
- 100 means exactly the same as the reference.
2. When scoring, consider layout, color scheme, typography, spacing, and element details.
3. Briefly explain the reason for each score.
4. Provide a final conclusion: which image is closer to the reference. The conclusion should be wrapped in LaTeX `\\boxed{}`.

---

### F.1. Human Evaluation

To ensure a stable and fair comparison, we recruit two independent human annotators, each tasked with evaluating every sample generated by UI2Code[N] against outputs from both closed- and open-source SOTA baselines, including Gemini-2.5-Pro, GPT-5, Claude-4-Sonnet, and Qwen2.5-VL-72B. Annotators assess multiple dimensions of quality, including visual structure and alignment, color and aesthetic design, and textual and content consistency.

Table 9 presents the full human evaluation statistics, including win, tie, and loss rates for UI2Code[N] against both open-source and closed-source baselines. Results are averaged over two independent annotators following a controlled pairwise comparison protocol. We additionally report the aggregated win-or-tie rate (Win+Tie) to reflect overall preference trends.

Table 10 reports the corresponding results produced by the VLM-based judge using the same pairwise comparison setting. Notably, the relative rankings and win-or-tie rates closely match those observed in human evaluation, indicating strong alignment between automated judgments and human preferences on challenging UI-to-code cases.

*Table 9.* Human evaluation results on the Design2Code-HARD benchmark. Values are averaged over two independent annotators.

| Comparison | Win | Tie | Lose | Win+Tie |
|---|---|---|---|---|
| Ours vs. GPT-5 | 56.3% | 16.3% | 27.5% | 72.5% |
| Ours vs. Gemini-2.5-Pro | 52.5% | 12.5% | 35.0% | 65.0% |
| Ours vs. Claude-4-Sonnet | 63.8% | 16.3% | 20.0% | 80.0% |
| Ours vs. Qwen2.5-VL-72B | 96.3% | 2.5% | 1.3% | 98.8% |

*Table 10.* VLM-based judge results on the Design2Code-HARD benchmark, using pairwise score comparison.

| Comparison | Win | Tie | Lose | Win+Tie |
|---|---|---|---|---|
| Ours vs. GPT-5 | 53.8% | 16.3% | 30.0% | 70.0% |
| Ours vs. Gemini-2.5-Pro | 40.0% | 21.3% | 38.8% | 63.8% |
| Ours vs. Claude-4-Sonnet | 63.8% | 17.5% | 18.8% | 81.3% |
| Ours vs. Qwen2.5-VL-72B | 91.3% | 5.0% | 3.8% | 96.3% |

### F.2. Human–VLM Agreement

To address concerns regarding the reliability and human alignment of our VLM-based evaluator, we conduct two complementary Human–VLM agreement studies on the Design2Code-HARD benchmark. The two studies examine alignment at both the **model-decision level** and the **sample-score level**, providing a comprehensive assessment of evaluator behavior.

Across both analyses, our evaluator demonstrates strong alignment with human judgments, conservative preference behavior, and stable, interpretable score calibration. These properties collectively indicate that the evaluator is reliable and suitable for UI-to-code evaluation and reinforcement learning.

**Model-Level Decision Alignment.** We first evaluate whether the VLM-based judge produces the same pairwise model comparison decisions as human annotators. For each of the 80 samples and each baseline model (GPT-5, Gemini-2.5-Pro, Claude-4-Sonnet, and Qwen2.5-VL-72B), two independent expert annotators and the VLM judge separately assess model outputs. Pairwise comparisons of *Ours vs. Baseline* are derived, yielding win, tie, and loss counts. Human decisions are aggregated by averaging the two annotators.

*Table 11.* Comparison of net-win margins between human evaluation and the VLM-based judge on the Design2Code-HARD benchmark.

| Baseline | Human-Avg Margin | VLM Margin |
|---|---|---|
| GPT-5 | 28.80% | 23.80% |
| Gemini-2.5-Pro | 17.40% | 1.20% |
| Claude-4-Sonnet | 43.50% | 45.00% |
| Qwen2.5-VL-72B | 95.00% | 87.50% |

We compute the net-win margin for each baseline as $(\text{Win} - \text{Loss})/N$. Table 11 reports the detailed statistics. Across all baselines, the two sets of margins exhibit strong agreement, with a Pearson correlation of 0.93 and a Spearman rank correlation of 1.0. Importantly, the VLM judge exactly reproduces the human ranking of baseline difficulty (Qwen2.5-VL-72B $\ll$ Claude-4-Sonnet $<$ GPT-5 $<$ Gemini-2.5-Pro), while assigning consistently more conservative preference magnitudes.

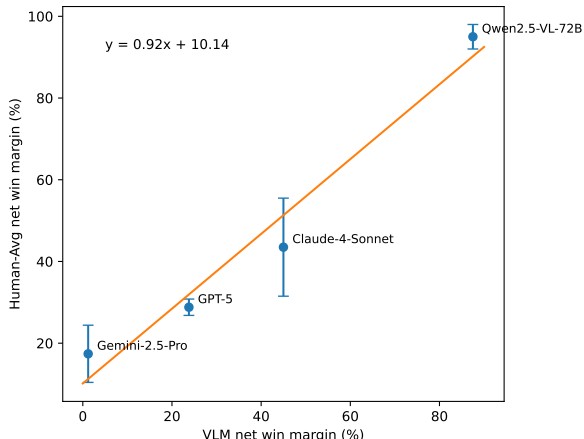

*Figure 6.* Correlation between VLM-based and human-averaged net-win margins across baselines on the Design2Code-HARD benchmark.

Figure 6 provides a visual comparison between human-averaged and VLM-based margins across baselines. Vertical error bars indicate human annotation variance, and the fitted regression line highlights the strong linear correspondence between the two. The monotonic ordering is perfectly preserved, further confirming reliable decision-level alignment between the VLM judge and human preferences.

Overall, for all baselines, both humans and the VLM judge agree on the preference direction (UI2Code[N] outperforming the baseline). The VLM judge consistently produces more conservative margins than humans, particularly for the strongest baseline (Gemini-2.5-Pro), indicating no bias toward our model and mitigating concerns about evaluator favoritism or reward gaming.

**Sample-Level Score Calibration.** We further analyze fine-grained score alignment between humans and the VLM evaluator at the per-sample level. Using the same 80 samples, three human annotators assign quality scores in the range $[0, 5]$ to outputs generated by UI2Code[N], while the VLM evaluator produces scores in $[0, 100]$.

Human judgments exhibit non-negligible subjectivity, with a per-sample standard deviation of 0.233, making the mean human score a standard and appropriate reference. At the score level, we observe moderate-to-strong alignment between the evaluator and human perception, with a Pearson correlation of 0.65 and a Spearman correlation of 0.44. These correlation magnitudes are consistent with those reported for widely adopted evaluators in subjective generative tasks, including RLAIF-based evaluators (Lee et al., 2023), ImageReward (Xu et al., 2023), HPS v2 (Wu et al., 2023), and Omni-Reward (Jin et al., 2025).

To assess discriminative power, we partition samples using a VLM score threshold of 80. Human scores for samples with VLM $\geq$ 80 are substantially higher (3.80–3.81) than

Table 12. Variance-aware evaluation on the Design2Code benchmark. Each model is evaluated across five independent runs, including both UI generation and VLM-based evaluation. Mean and standard deviation are reported.

| Model | R1 | R2 | R3 | R4 | R5 | Mean | Std |
|---|---|---|---|---|---|---|---|
| Qwen2.5-VL-72B | 41.9 | 41.6 | 39.5 | 40.8 | 38.9 | 40.54 | 1.31 |
| GPT-5 | 89.7 | 89.3 | 89.5 | 90.2 | 90.6 | 89.86 | 0.53 |
| UI2Code$^N$-SFT | 79.3 | 79.0 | 78.8 | 79.9 | 78.7 | 79.14 | 0.48 |
| UI2Code$^N$-RL | 88.6 | 88.5 | 88.2 | 87.7 | 88.4 | 88.28 | 0.36 |

those with VLM $< 80$ (2.50–2.80), corresponding to a large effect size (Cohen's $d = 1.32$–$1.82$). This indicates that the evaluator effectively separates high- and low-quality outputs.

Finally, we evaluate binary agreement by treating human scores $\geq 4$ as high-quality outputs. Under this criterion, the VLM evaluator achieves 82.3% agreement, 98% recall, and approximately 0.82 precision, with Cohen's $\kappa = 0.41$. The high recall suggests that the evaluator rarely rejects outputs preferred by humans, a desirable property for both evaluation and reward modeling.

Overall, these results demonstrate that the proposed evaluator exhibits reliable Human–VLM alignment at the sample level, with interpretable calibration and conservative behavior comparable to existing multimodal evaluators.

### F.3. Variance and Stability Analysis

We further analyze the stability of VLM-based evaluation by measuring score variance across repeated runs and different evaluator instances. Specifically, we re-run the full evaluation pipeline five times, including both (i) the UI generation process of each evaluated model and (ii) the VLM-based evaluation procedure. We conduct experiments on four representative models: Qwen2.5-VL-72B, GPT-5, and the SFT and RL variants of our proposed UI2Code$^N$ (9B). For each run, models are evaluated independently under the same benchmark and evaluation protocol.

Table 12 summarizes the results across five independent runs. Across all runs, the relative ranking of models remains identical, indicating that the evaluation pipeline is highly stable and not sensitive to stochastic variation. Moreover, all models exhibit low variance across runs, with the only exception being Qwen2.5-VL-72B, whose substantially lower performance naturally leads to greater fluctuation.

Notably, UI2Code$^N$-RL achieves performance comparable to the significantly larger commercial model GPT-5 while exhibiting the lowest standard deviation among all evaluated models. This consistent performance across repeated runs further supports the robustness and reliability of our method, and confirms that the reported improvements are not artifacts of random variation.

## G. Additional Experiments

We conduct a series of additional experiments to further analyze and validate the proposed UI2Code$^N$ model.

**Impact of the *Think/Answer* Format.** We evaluate the effect of the *Think/Answer* generation format by comparing it against a *Direct Answer* baseline on the Design2Code benchmark. As shown in Table 13, incorporating an explicit *Think* stage yields a consistent performance improvement of +3.1%. Qualitatively, the *Think* stage enables the model to perform high-level structural reasoning prior to code generation, such as identifying major layout regions (e.g., header and main content) and planning the corresponding DOM hierarchy. This form of visual planning helps reduce layout hallucination and leads to more structurally coherent HTML generation.

Table 13. Impact of the *Think/Answer* format on the Design2Code benchmark.

| Thinking Mode | Design2Code |
|---|---|
| With *Think/Answer* | 88.6 |
| Without *Think* | 85.5 |

**Cross-Verification with Held-out Evaluators.** To further validate the reliability and calibration of our VLM-based judge, we perform cross-verification using a diverse set of held-out evaluation metrics on the Design2Code benchmark. These metrics include traditional element-matching measures proposed in Design2Code, namely *Block*, *Text*, *Position*, and *Color*, as well as vision feature–based similarity metrics, including CLIP similarity (ViT-B-32) and DINOv3 similarity (facebook/dinov3-vitl16-pretrain-lvd1689m). All metrics are computed on rendered webpages under the same evaluation protocol. Table 14 reports the results across different models. Across all metrics, model rankings are largely consistent: GPT-5, Gemini-2.5-Pro, and UI2Code$^N$ form the top-performing tier, while Qwen2.5-VL-72B consistently ranks lowest. Notably, although traditional element-matching and vision feature similarity metrics capture coarse visual alignment, they exhibit limited discriminative power among top-performing models. In contrast, the VLM-based judge produces clearer and more calibrated performance gaps, whose magnitudes align more closely with human evaluation. This suggests that the VLM judge more faithfully captures human-perceived UI quality by jointly assessing layout structure, visual semantics, and rendered appearance, rather than relying on isolated HTML attributes or embedding-level similarity alone.

**Additional Comparison with Specialized UI-to-Code Models.** To complement the comparisons with general-purpose VLMs in the main experiments, we additionally compare UI2Code$^N$ with representative specialized UI-to-

*Table 14.* Cross-verification of UI-to-code performance on the Design2Code benchmark using traditional element-matching metrics, vision feature similarity metrics, and the VLM-based judge.

| Model | Block | Text | Position | Color | CLIP | DINOv3 | VLM Judge |
|---|---|---|---|---|---|---|---|
| GPT-5 | 89.1 | 94.2 | 86.4 | 78.0 | 81.6 | 87.7 | 89.7 |
| Gemini-2.5-Pro | 89.1 | 93.5 | 85.5 | 71.4 | 80.9 | 85.6 | 89.5 |
| Claude-4-Sonnet | 88.7 | 93.2 | 84.6 | 72.0 | 80.5 | 85.5 | 81.2 |
| Qwen2.5-VL-72B | 86.6 | 91.6 | 76.8 | 67.8 | 77.8 | 74.4 | 41.9 |
| UI2Code$^N$-SFT | 86.8 | 91.5 | 81.7 | 69.7 | 79.0 | 78.8 | 79.3 |
| UI2Code$^N$-RL | 88.7 | 93.1 | 83.8 | 72.6 | 80.5 | 86.1 | 88.6 |

code models and systems. We conduct this comparison on WebCode2M-Long, following the evaluation protocol used by prior UI-to-code work and reporting CLIP similarity as the evaluation metric. As shown in Table 15, UI2Code$^N$-RL outperforms WebSight VLM-7B, Design2Code-18B, and UICopilot. These results provide additional evidence that UI2Code$^N$ is competitive not only with general-purpose VLMs, but also with specialized UI-to-code models.

*Table 15.* Additional comparison with specialized UI-to-code models on the WebCode2M-Long benchmark.

| Model | CLIP Similarity |
|---|---|
| WebSight VLM-7B | $0.69 \pm 0.12$ |
| Design2Code-18B | $0.74 \pm 0.10$ |
| UICopilot | $0.77 \pm 0.11$ |
| UI2Code$^N$-RL | **$0.79 \pm 0.09$** |

**Oscillations in the UI Polishing Process.** While the overall polishing performance exhibits a stable and positive trend across rounds (as shown in Table 2 of the main paper), we observe occasional oscillations and quality regressions at the individual-sample level. To better understand this behavior, we conduct a fine-grained analysis from both macroscopic and microscopic perspectives.

**Macroscopic Stability.** Table 16 reports the average polishing accuracy across multiple refinement rounds. Although minor fluctuations are observed, the overall trend remains stable, indicating that the iterative refinement process is globally effective rather than divergent.

**Microscopic Oscillations.** To investigate the root cause of individual-level fluctuations, we stratify samples based on their initial UI2Code score using 80 as a convenient split point. This threshold is used purely for analysis and does not imply an intrinsic difficulty boundary. We then compute the *Polish Success Rate*, defined as the probability that a polishing step improves the score relative to the previous round.

As shown in Table 17, for lower-quality cases (initial score < 80), the model achieves a high success rate of 74.5%, demonstrating strong effectiveness in correcting substantive errors such as layout structure issues or missing components.

*Table 16.* Average polishing accuracy across successive refinement rounds.

| Polish Round | Polish Accuracy (%) |
|---|---|
| 1 | 66.0 |
| 2 | 64.7 |
| 3 | 63.3 |
| 4 | 65.8 |

*Table 17.* Polish success rate stratified by initial UI2Code score.

| Initial UI2Code Score Range | Polish Success Rate (%) |
|---|---|
| $\geq 80$ | 53.1 |
| $< 80$ | 74.5 |

In contrast, for higher-quality cases (initial score $\geq 80$), the success rate decreases to 53.1%, indicating that the model has largely reached its capability ceiling. In this regime, further polishing yields diminishing returns, and observed oscillations often correspond to stylistic variations rather than meaningful functional improvements.

Overall, these results indicate that polishing oscillations primarily arise near convergence and do not undermine the stability or effectiveness of the iterative refinement process.

**Evaluator Sensitivity and Stability.** To assess the sensitivity and robustness of our evaluation results with respect to evaluator choice, we conduct a comprehensive stability analysis across four diverse VLM-based evaluators: GPT-o4-mini, GLM-4.5V, Claude-4-Sonnet, and Gemini-2.5-Pro. These evaluators differ substantially in architecture, training data, and scale, providing a strong testbed for evaluator invariance. Across all evaluators, we observe perfect rank-order consistency among the compared models: GPT-5 > UI2Code$^N$-RL > UI2Code$^N$-SFT > Qwen2.5-VL-72B. Moreover, score trends are highly correlated across evaluators, with Pearson correlation coefficients exceeding 0.98. Importantly, relative performance gaps remain stable: GPT-5 and UI2Code$^N$-RL consistently form the top tier with the smallest gap between them, while Qwen2.5-VL-72B exhibits the largest margin to all other models. In addition, UI2Code$^N$-RL consistently outperforms its SFT counterpart by 6–10 points across all evaluators. This demonstrates that

the gains introduced by reinforcement learning are evaluator-invariant rather than being tied to any specific judge. Overall, these results confirm that our conclusions are robust to evaluator choice and that the observed improvements reflect genuine model capability rather than evaluator bias.

**Effect of Evaluator Choice.** We employ different VLMs as evaluators for UI drafting and UI polishing, based on the distinct capability requirements of the two tasks and empirical human-alignment studies. This choice is not arbitrary, but guided by systematic validation of evaluator reliability, alignment, and cost–performance trade-offs.

**Task-Specific Requirements.** UI drafting involves a direct **pairwise** comparison between a reference screenshot and a rendered webpage, focusing on overall visual similarity. In contrast, UI polishing requires a more challenging **triplet** comparison among a reference screenshot, an initial rendering, and a polished rendering, where the evaluator must determine whether the refined output represents a relative improvement toward the reference. This setting demands finer-grained visual discrimination and more reliable relative judgment.

**Evaluator Selection via Human Alignment.** For UI-to-code generation, we adopt GPT-o4-mini as the evaluator. We validate its reliability through a human study on 100 randomly sampled Design2Code cases, where GPT-o4-mini achieves 92% agreement with human judgments. Moreover, its scores exhibit an extremely high correlation with Gemini-2.5-Pro on this task ($r = 0.9998$), indicating that stronger models do not materially change evaluation outcomes. Given its substantially lower cost, GPT-o4-mini provides an effective and efficient choice for pairwise similarity evaluation.

For UI polishing, we find that GPT-o4-mini is insufficient for detecting subtle visual improvements in the triplet setting. Qualitative inspection reveals frequent failures in distinguishing which rendering is closer to the reference. We therefore conduct a pilot human-alignment study comparing multiple VLMs. As summarized in Table 19, Gemini-2.5-Pro achieves the highest agreement with human preferences (94%), substantially outperforming GPT-o4-mini (77%) and GPT-5 (85%). We thus adopt Gemini-2.5-Pro as the evaluator for UI polishing, where accurate relative judgment is critical.

Overall, these results demonstrate that our evaluator choices are task-driven and empirically justified, balancing human alignment and computational efficiency while ensuring reliable evaluation across different UI-to-code settings.

**Efficiency Analysis of RVPO.** We provide an empirical efficiency analysis of the pairwise comparison step in Relative Visual Policy Optimization (RVPO). Although RVPO defines relative preferences over candidate pairs, the pairwise comparison step is not the computational bottleneck in practice. The dominant cost comes from autoregressive generation of rollout candidates, while VLM-based pairwise scoring is performed as a post-hoc comparison over already generated outputs.

In our implementation, candidate rollouts are generated once and reused for relative visual comparison. Therefore, the cost of pairwise comparison is decoupled from the expensive token generation process. We further use a tournament-style aggregation strategy to reduce unnecessary comparisons while preserving a stable relative preference signal. Table 20 shows wall-clock time breakdown during RVPO training. These results show that RVPO remains computationally practical: the pairwise comparison stage contributes only a minor fraction of the overall training time, while the majority of the cost is dominated by rollout generation and rendering.

## H. Demo Cases

To provide an intuitive understanding of the proposed UI2Code$^N$, we present several representative demo cases focusing on UI-to-code and UI Editing:

- **UI-to-Code**: Given a raw UI screenshot, the model automatically generates executable HTML/CSS code that faithfully reproduces the layout, color scheme, and visual elements of the design. The demos show that our model is able to handle both simple layouts and complex, nested structures with high fidelity.

- **UI Editing**: Starting from an existing rendering, the model is able to perform targeted edits such as modifying layout alignment, adjusting typography, changing color themes, or inserting new components. These cases demonstrate the model's ability to act as an interactive assistant in iterative design workflows.

These demo cases highlight the versatility of our system across different aspects of UI development, demonstrating its potential as both a code generator and an interactive design assistant.

### H.1. Cases of UI2Code

The following examples illustrate the UI-to-code capability of UI2Code$^N$ across a range of layout complexities. These cases include both relatively simple single-section designs and more complex webpages with nested structures, long vertical layouts, and diverse visual components. They demonstrate that the model can faithfully reconstruct global layout, hierarchical structure, and visual style directly from raw UI screenshots.

*Table 18.* Evaluation results across different VLM-based evaluators, showing score stability and consistent ranking.

| Model | o4-mini | | GLM-4.5V | | Claude-4-Sonnet | | Gemini-2.5-Pro | |
|---|---|---|---|---|---|---|---|---|
| | Score ↑ | Rank | Score ↑ | Rank | Score ↑ | Rank | Score ↑ | Rank |
| Qwen2.5-VL-72B | 41.9 | 4 | 56.6 | 4 | 33.5 | 4 | 33.1 | 4 |
| GPT-5 | 89.7 | 1 | 93.2 | 1 | 84.7 | 1 | 82.6 | 1 |
| UI2Code$^N$-SFT | 79.3 | 3 | 84.7 | 3 | 75.4 | 3 | 72.7 | 3 |
| UI2Code$^N$-RL | 88.6 | 2 | 91.1 | 2 | 84.5 | 2 | 82.2 | 2 |

*Table 19.* Human agreement rates of different VLM evaluators on the UI polishing task, illustrating the increased difficulty of triplet-based relative comparison.

| Evaluator | Human Agreement (UI Polishing) |
|---|---|
| GPT-o4-mini | 77% |
| GPT-5 | 85% |
| Gemini-2.5-Pro | 94% |

*Table 20.* Wall-clock time breakdown during RVPO training.

| Stage | Time (s) | Proportion (%) |
|---|---|---|
| Generation | 130.9 | 75.4 |
| Rendering | 38.8 | 22.4 |
| Comparison | 3.9 | 2.2 |

## H.2. Cases of UI Editing

We further present representative UI editing cases, where UI2Code$^N$ performs targeted modifications on existing renderings. These examples demonstrate the model's ability to follow localized editing instructions, such as adjusting layout alignment, updating typography, changing color themes, and inserting new components. The results highlight the model's suitability as an interactive assistant in iterative UI design workflows.

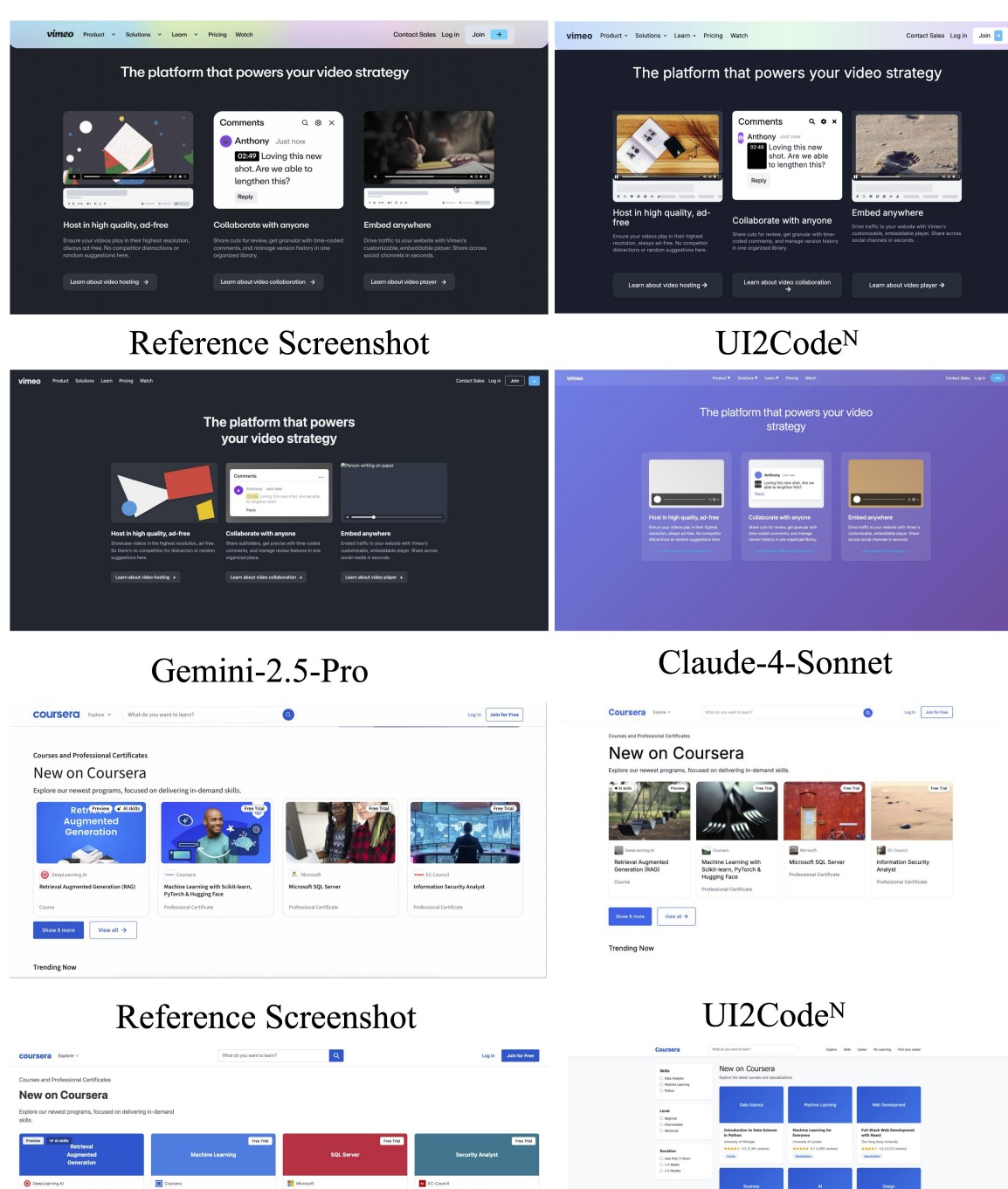

Reference Screenshot

UI2Code$^N$

Gemini-2.5-Pro

Claude-4-Sonnet

Reference Screenshot

UI2Code$^N$

Gemini-2.5-Pro

Claude-4-Sonnet

*Figure 7.* UI2Code$^N$ Demo Cases: UI-to-code (1/4)

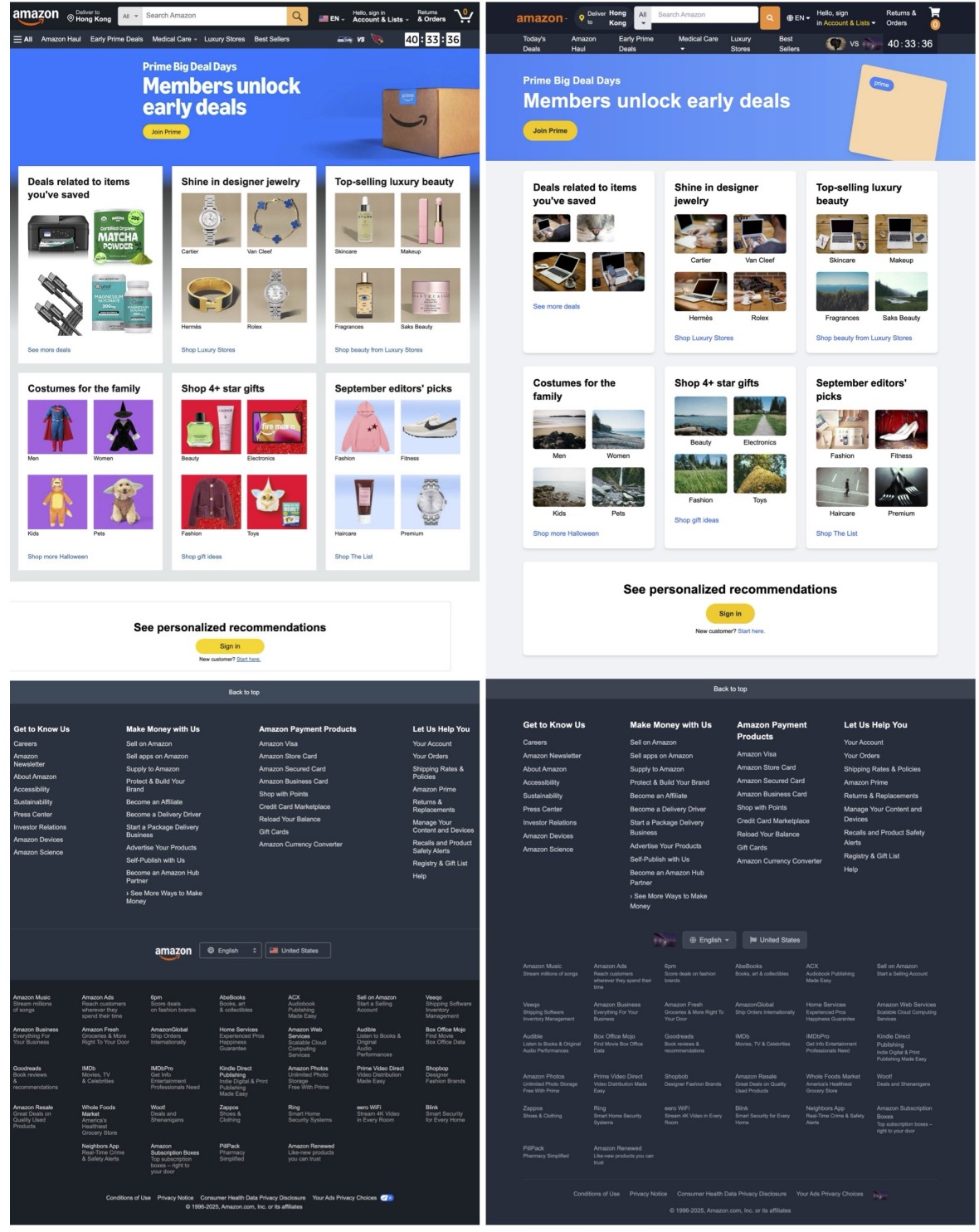

*Figure 8.* UI2Code$^N$ Demo Cases: UI-to-code (2/4)

## Reference Screenshot

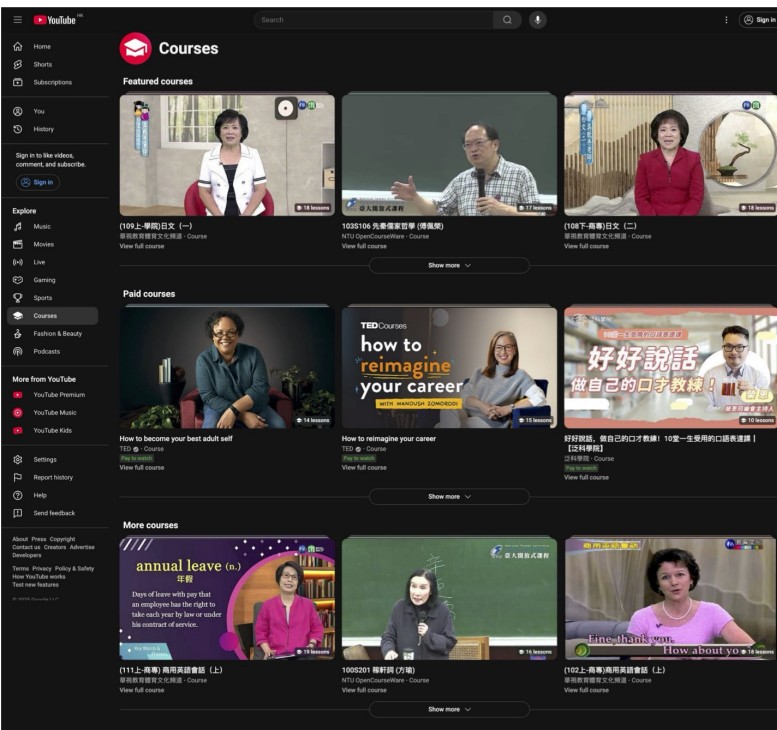

## UI2Code$^N$

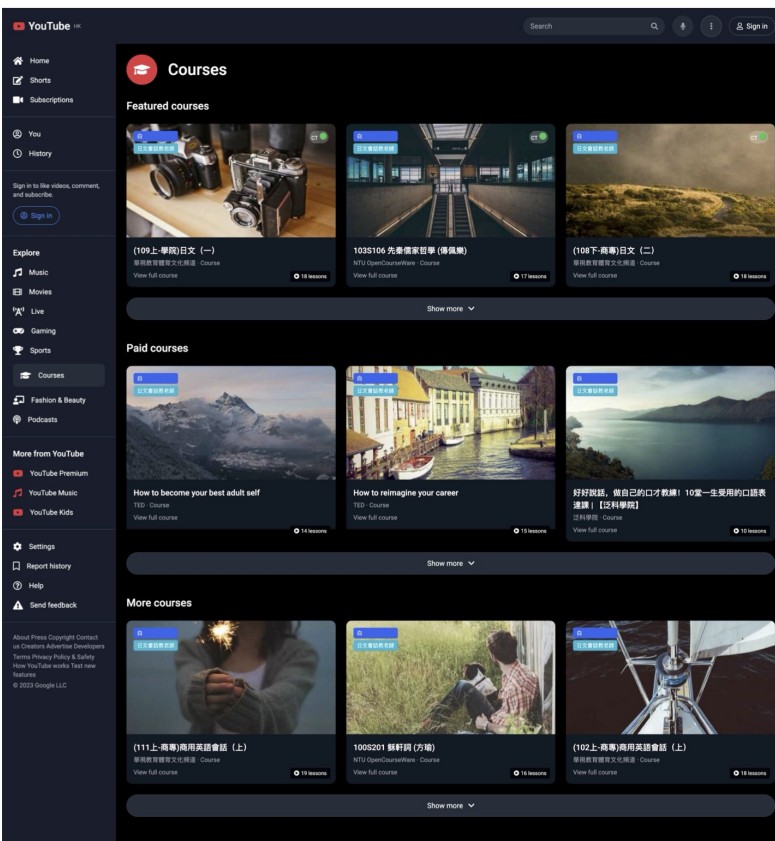

*Figure 9.* UI2Code$^N$ Demo Cases: UI-to-code (3/4)

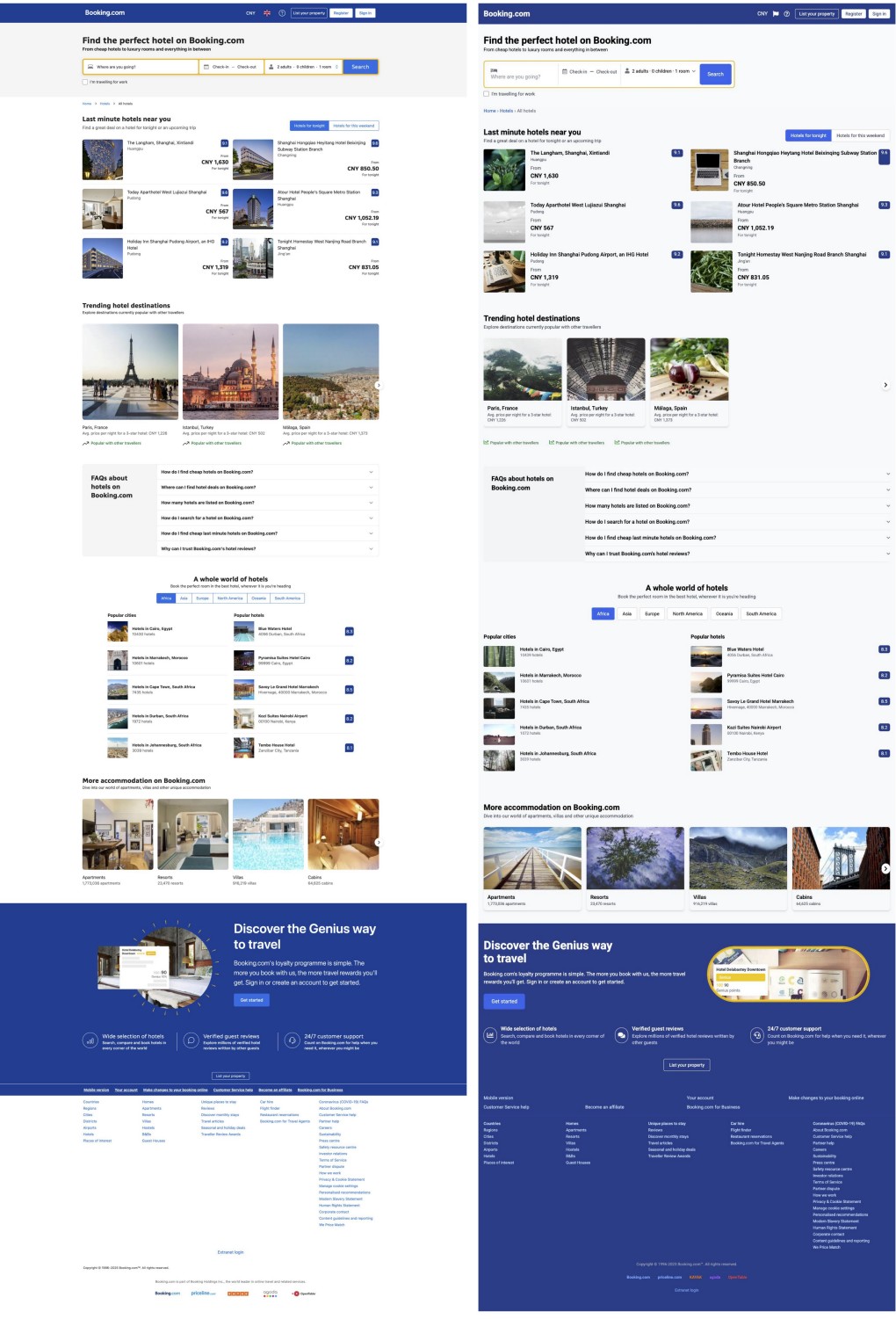

*Figure 10.* UI2Code$^N$ Demo Cases: UI-to-code (4/4)

UI Editing: Change the news on the top to a Christmas story.

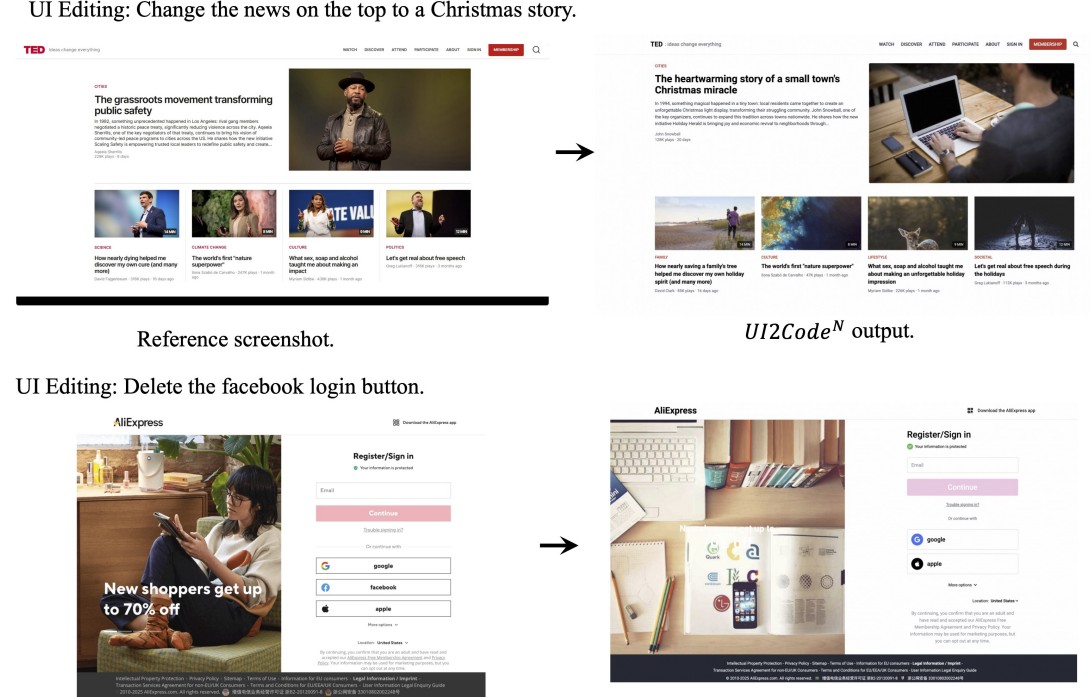

Reference screenshot.                                   $UI2Code^N$ output.

UI Editing: Delete the facebook login button.

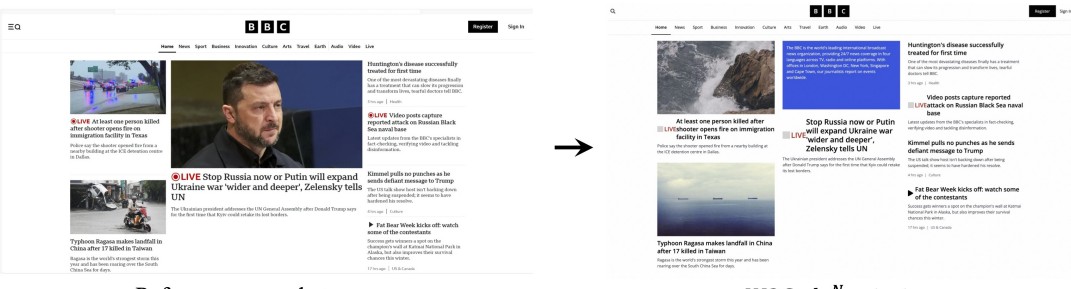

Reference screenshot.                                   $UI2Code^N$ output.

UI Editing: Replace the photo of Zelensky with a short paragraph describing BBC on a blue background.

Reference screenshot.                                   $UI2Code^N$ output.

UI Editing: Align the 'New on Coursera' texts to the right side.

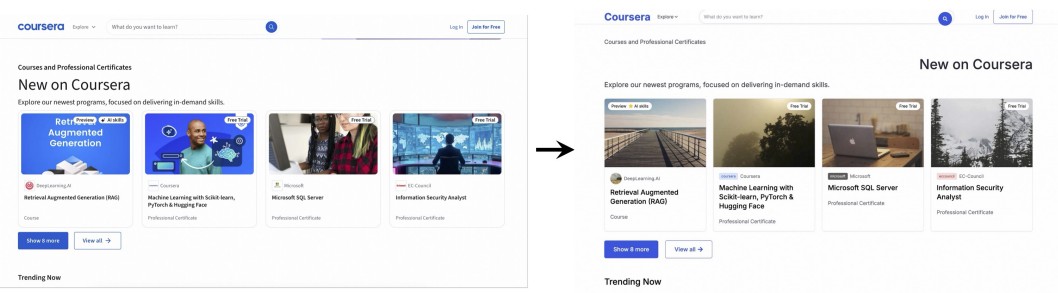

Reference screenshot.                                   $UI2Code^N$ output.

*Figure 11.* UI2Code$^N$ Demo Cases: UI Editing (1/2)

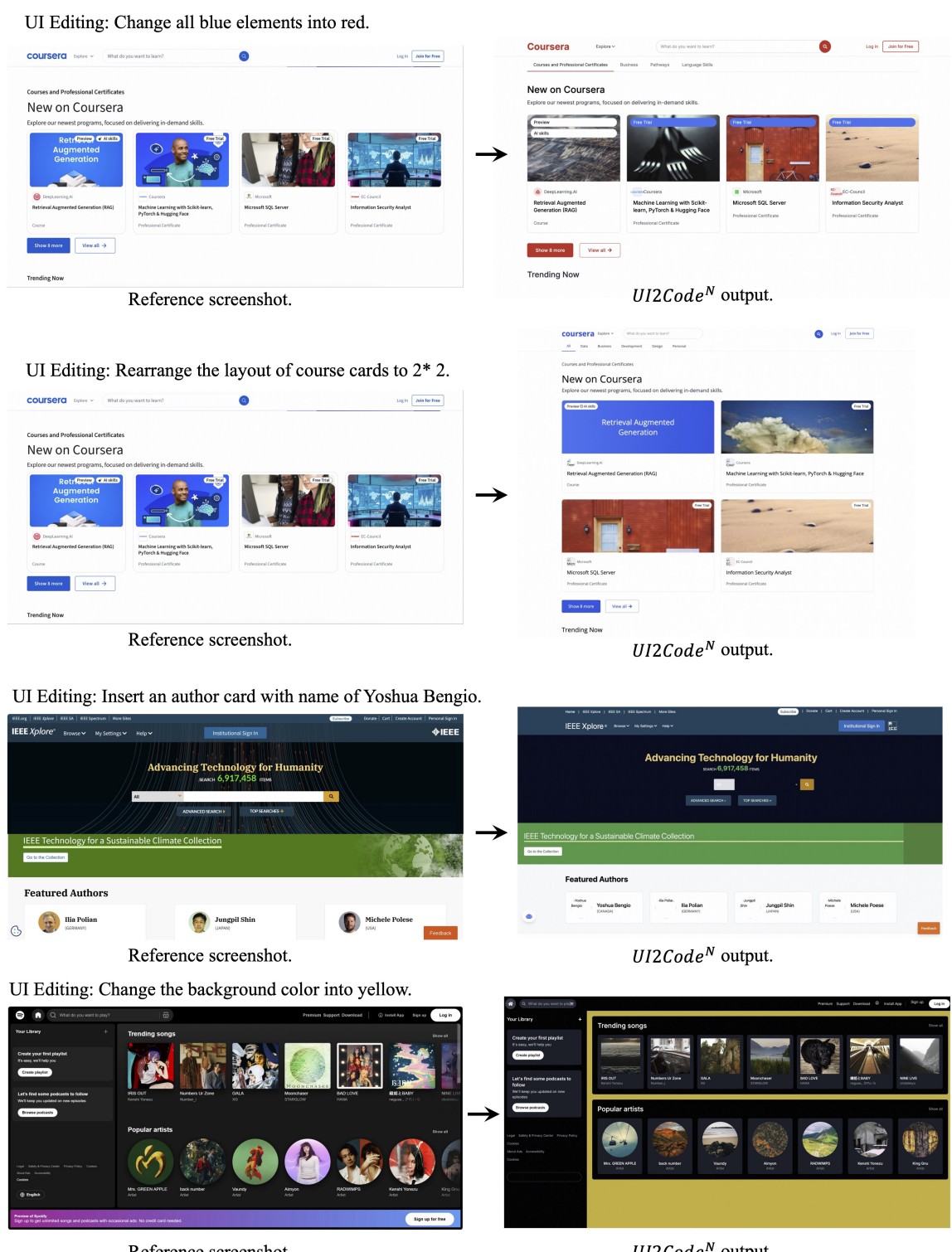

*Figure 12.* UI2Code$^N$ Demo Cases: UI Editing (2/2)

