# OpenReview forum: "UI2Code^N: UI-to-Code Generation as Interactive Visual Optimization"
_ICML.cc/2026/Conference — ICML 2026 regular_

### Official Review · Reviewer_GpZw · 2026-03-10

**Soundness:** 3
**Presentation:** 3
**Significance:** 2
**Originality:** 3
**Overall Recommendation:** 4
**Confidence:** 3

**Summary:**

Existing UI-to-code models usually generate code in a single pass, which ignores the iterative nature of real frontend development. To fix this, the authors treat the task as an interactive visual optimization problem and propose Relative Visual Policy Optimization (RVPO). Instead of using noisy absolute scores, RVPO ranks multiple rendered candidates to update the model. Based on this method, they trained a 9B open-source model named UI2Code$^N$. It achieves state-of-the-art results on UI drafting, polishing, and editing tasks, beating much larger closed-source models. Furthermore, experiments show that more test-time iterations consistently improve the visual quality of the generated code.

**Compliance With Llm Reviewing Policy:**

Affirmed.

**Final Justification:**

I keep my original weak-accept stance

**Key Questions For Authors:**

1.  Given the strict non-release policy for your datasets and benchmarks, what specific mechanisms (e.g., open-sourcing data-curation scripts or releasing heavily anonymized subsets) will you provide to allow the community to independently verify your claims and audit ethical compliance?
2. Since RVPO relies entirely on visual feedback, how do you prevent "reward hacking" where the model generates visually accurate but structurally poor code (e.g., overusing absolute positioning instead of scalable Flexbox/Grid layouts)?
3.  How does the model manage context during the multi-round polishing process to retain historical edits without causing a context window explosion from stacking multiple rendered images and long HTML sequences?
4. How does the visual judger isolate genuine structural/CSS errors from visual discrepancies caused purely by missing external assets (e.g., broken image links or CORS issues during rendering)?

**Limitations:**

Yes.

**Strengths And Weaknesses:**

The key problem assessed by this paper is that existing UI-to-code models treat the task as a single-turn generation, ignoring the iterative, feedback-driven reality of frontend development. This study's principal contribution is reconceptualizing UI-to-code as an interactive visual optimization problem and introducing Relative Visual Policy Optimization (RVPO) to train a 9B model (UI2Code$^N$) using non-differentiable rendering feedback.

Originality and Significance: The shift from single-pass generation to a closed-loop, test-time scalable process is highly original and aligns perfectly with real-world engineering. Unifying UI drafting, polishing, and editing under one framework significantly advances VLM utility in frontend automation. Additionally, using RVPO to bypass the instability of absolute VLM rewards is a clever, impactful solution to a known bottleneck in visual reinforcement learning.

Soundness & Reproducibility:  While the methodology is technically robust and ablation studies clearly validate RVPO, the paper suffers from severe reproducibility and transparency bottlenecks. The strict non-release policy for the 10M pretraining corpus and the new benchmarks (`UI2Code-Real`, `UIPolish-bench`) creates a closed ecosystem. Because the pipeline also relies on proprietary models (like Gemini-2.5-Pro) for reward data distillation, the community cannot independently verify the benchmark results, stress-test the claims, or audit the ethical compliance of the scraped data. Furthermore, the structural quality of the generated code is underexplored—purely visual rewards might encourage "reward hacking," such as overusing absolute positioning instead of scalable layouts.

Presentation:  The paper is exceptionally well-written and logically structured. The extensive appendices provide crucial details on prompts, human-VLM agreement, and qualitative cases, effectively defending the robustness of the VLM evaluators.

---

> ### Author Rebuttal · Authors · 2026-03-31
>
> We sincerely thank the reviewer for the valuable feedback. We will address each point one by one as follows:
>
> ### **Response to Q1: Addressing Transparency and Reproducibility Concerns**
>
> We thank the reviewer for raising this important concern. While our datasets and benchmarks are not fully released, we are committed to ensuring transparency and reproducibility. To address this, we plan to implement the following mechanisms:
>
> **(1) Open-Sourcing Benchmarks and Models**: We will release the benchmarks (including **UI2Code-Real** and **UIPolish-bench**) and the trained models for public use. These resources will enable the community to verify our results and extend our work.
>
> **(2) Open-Sourcing Evaluation Pipeline**: To ensure reproducibility, we will open-source our evaluation pipeline, including the **inference script**, **judge script**, and **score calculation procedures**. This will enable the community to independently verify and replicate the experiments.
>
> **(3) Releasing SFT Data Subset**: We will release a heavily anonymized subset of the **SFT data**, which includes **UI screenshots** and their corresponding code. This subset will allow for independent verification while maintaining privacy.
>
> Our **Ethical Compliance** procedures, including adherence to **robots.txt**, privacy protection, and licensing constraints, are outlined in the **Ethics Statement** section of the manuscript.
>
> By releasing these resources and clarifying the data processing pipeline, we aim to foster transparency and allow the community to independently verify and audit our work.
>
> ---
>
> ### **Response to Q2:  Preventing "Reward Hacking" in RVPO**
>
> While **RVPO** relies on visual feedback, we have designed the system with specific mechanisms to prevent reward hacking.
>
> Our approach uses a **prompt-driven visual judger** that evaluates **layout structure**, **color fidelity**, **typography**, **spacing**, and **element details**, encouraging the model to prioritize **responsive design** and **scalable layouts** over visual similarity.
>
> The **visual judger**, working in conjunction with **RVPO**, ranks multiple candidates based on **relative visual preferences**. This ranking system compares and ranks different candidate outputs relative to each other, which reduces the potential for the model to generate visually accurate but structurally poor code (such as over-relying on absolute positioning). This **multi-candidate ranking mechanism** ensures that the model is encouraged to generate not only visually accurate code but also structurally sound, as it optimizes across a range of potential solutions rather than merely focusing on a single output.
>
> We quantitatively verify that **UI2Code^N** does not suffer from reward hacking. We parsed the generated **HTMLs** on the **Design2Code** benchmark and calculated the ratio of **hard-coded absolute positioning**. The results show that **UI2Code^N-RL** continues to generate well-structured, responsive code, with a **low** percentage of **absolute positioning**.
>
> | Model              | Absolute Positioning Ratio |
> |--------------------|----------------------------|
> | **UI2Code^N-SFT**    | 0.7%                         |
> | **UI2Code^N-RL**     | 0.5%                         |
>
> ---
>
> ### **Response to Q3: Managing Context in Multi-Round Polishing**
> In multi-round polishing, we keep only the latest polishing rounds rather than the full history. When the context limit is reached, earlier intermediate rounds are removed. This is reasonable because polishing is a progressive refinement process: later-round outputs typically incorporate and improve upon earlier edits, making the most recent result the most informative state for subsequent polishing. In practice, this design remained effective while keeping context growth manageable.
>
> ---
>
> ### **Response to Q4: Addressing Visual Judger and Missing External Assets**
>
> Thank you for highlighting this important challenge in UI-to-code generation. To explicitly address this issue, our visual judger uses a prompt-driven evaluation scheme that focuses on structural and stylistic fidelity between the reference image and the rendered output, covering aspects such as:
>
> - **Layout structure** (positions, alignment, overall layout),
> - **Color fidelity**,
> - **Typography**,
> - **Spacing ratios**,
> - **Element details**.
>
> The evaluation is designed to **ignore discrepancies** caused by missing external assets, such as broken image links or CORS issues, since the prompt emphasizes structural correctness and style fidelity rather than the presence of externally hosted resources. In addition, based on our manual inspection, such missing assets typically do not materially affect the judgment of layout and styling consistency in practice.
>
> Overall, this prompt-driven design allows the judger to focus on layout and structure while remaining largely robust to missing external assets.

---

> > ### Author Rebuttal · Reviewer_GpZw · 2026-04-03
> >
> > My concerns have been adequately addressed.

---

> > > ### Author Response · Authors · 2026-04-08
> > >
> > > Dear Reviewer,
> > >
> > > Thank you for your time and the constructive feedback. We are very pleased that our responses and additional experiments have fully resolved the concerns raised.
> > >
> > > We will incorporate all discussed clarifications into the final version and reiterate our commitment to open-sourcing the UI2Code^N models and benchmarks. Thank you again for recognizing the value of UI2Code^N.
> > >
> > > Authors of UI2Code^N

---

### Official Review · Reviewer_GjW1 · 2026-03-12

**Soundness:** 3
**Presentation:** 3
**Significance:** 2
**Originality:** 3
**Overall Recommendation:** 4
**Confidence:** 4

**Summary:**

This paper tries to solve screenshot-to-code generation problem using an iterative refinement approach where each iteration the model takes the target screenshot, current code and the rendering of the current code, and outputs a new code that is more similar to the target screenshot.
It uses a 9B pre-trained VLM model and do continual pretraining on webpage image/code pairs, supervised fine-tuning for drafting, polishing, and editing.
After that there's an RL stage (RVPO) that samples multiple code candidates, renders them, judges them pairwise in visual space, and optimizes relative rewards with a GRPO-style objective.
The model is evaluated on a public benchmarks and also a new polishing benchmarks. plus human evaluation for hard drafting and editing cases.
The results suggests that RL and multi-round polishing improve the base model substantially and yield a practically strong open model, especially on the paper's own polishing and real-world evaluations.

**Compliance With Llm Reviewing Policy:**

Affirmed.

**Key Questions For Authors:**

* Are the 10M crawl-based CPT data and 80K SFT data deduplicated against Design2Code, Web2Code, Flame, UI2Code-Real, UIPolish-bench, and UIEdit-Bench. What's the process to prevent eval leaking?

* Were GPT-5/Gemini/Claude/Qwen given the same multi-round refinement opportunity and comparable test-time budgets as UI2CodeN?

* missing citations for the related works:
    * Rodriguez et al. Rendering-Aware Reinforcement Learning for Vector Graphics Generation
    * Gui et al. UICopilot: Automating UI Synthesis via Hierarchical Code Generation from Webpage Designs.

**Limitations:**

yes

**Strengths And Weaknesses:**

Strength

* The draft-render-polish loop is well motivated and a good approach for frontend generation, since many errors only become visible after execution, and this is similar to the way humans work.

* The RVPO approch is a good way to improve the model without creating intricate reward functions especially during late stage where absolute rewards starting to saturate and becomes unstable.

* good evaluation package and ablation studies.

Weakness

* The paper's claim of broad novelty and SOTA framing is stronger than the evidence provided. The interactive correction idea has close antecedents in recent UI-to-code/self-correction/editing work, and the RL component looks closer a variant of (Rodriguez et al. Rendering-Aware Reinforcement Learning for Vector Graphics Generation) than a fundamentally new optimization method. The comparison set is broad in model count but incomplete with respect to recent specialized UI-to-code systems and newer execution/editing benchmarks, so the paper does not yet establish leadership across the area.

---

> ### Author Rebuttal · Authors · 2026-03-31
>
> We thank the reviewer for the feedback. We will address each concern individually as follows.
>
> ### **Response to W1: Clarification on Novelty and Benchmark Comparison**
>
> While there are similarities with prior UI-to-code and self-correction work, our core contribution lies in reformulating UI-to-code as an **interactive visual optimization problem**, using **Relative Visual Policy Optimization (RVPO)** for iterative refinement based on executable feedback. This is a distinct departure from prior approaches.
>
> Regarding the **RL component**, unlike **Rodriguez et al.**, who optimize with a single reward signal derived from rendered output, our RVPO leverages **relative visual preferences** in multi-candidate optimization, providing a more stable and robust reward signal for visual tasks.
>
> We recognize the importance of a comprehensive comparison set. We perform a comparison of **UI2Code^N** with several baselines, including **WebSight VLM-7B**, **Design2Code-18B**, and **UICopilot**, on the **WebCode2M-Long benchmark** using the **CLIP** metric. The results indicate that **UI2Code^N-RL** performs significantly better than other models.
>
> | Model               | CLIP |
> |---------------------|------|
> | WebSight VLM-7B     | 0.69 (±0.12) |
> | Design2Code-18B               | 0.74 (±0.10) |
> | UICopilot                       | 0.77 (±0.11) |
> | UI2Code^N-RL                  | 0.79 (±0.09) |
>
> Besides, we have included a detailed comparison with **agent-based systems** on the **Design2Code-HARD benchmark** (Table 13 in Appendix), where **UI2Code^N-RL** demonstrates superior performance compared to models such as **DCGen** and **ScreenCoder** in accuracy, latency, and token cost.
>
> We will revise the paper to better emphasize these results and clarify the leadership of our approach in the field.
>
> ---
>
> ### **Response to Q1: Data decontamination and potential leakage**
>
> As described in our response to Reviewer ErBe's W3, we have performed **image-level deduplication** using techniques such as hash-based filtering to remove near-duplicate samples from the training set.
>
> ---
>
> ### **Response to Q2: Clarification on Multi-Round Refinement and Test-Time Budgets**
>
> In our experiments, **GPT-5**, **Gemini**, **Claude**, and **Qwen** were given the same **multi-round refinement opportunity** and comparable **test-time budgets** as **UI2Code^N**. All models were evaluated under identical conditions, including the same number of refinement rounds and test-time budgets. However, it's important to note that these models do not have the same **Polish capabilities** as **UI2Code^N**, which is specifically designed for iterative refinement of UI elements. As a result, while all models underwent the same number of refinement rounds, **UI2Code^N** benefits from a more specialized **polishing process**, as reflected in **Table 1**. We will clarify this distinction in the revised manuscript to highlight the unique strength of **UI2Code^N** in polish tasks.
>
> ---
>
> ### **Response to Q3: Missing Citations**
> We thank the reviewer for pointing out the missing citations. We will add the relevant references to the revised manuscript.

---

### Official Review · Reviewer_ErBe · 2026-03-14

**Soundness:** 3
**Presentation:** 3
**Significance:** 3
**Originality:** 3
**Overall Recommendation:** 4
**Confidence:** 4

**Summary:**

The paper introduces UI2CodeN, a 9B parameter vision-language model that reformulates UI-to-code generation from a single-pass task into an interactive visual optimization problem. By employing a novel preference-based reinforcement learning algorithm called Relative Visual Policy Optimization (RVPO), the model iteratively refines executable front-end code based on rendered visual feedback. The approach utilizes a three-stage training pipeline—continual pre-training, supervised fine-tuning, and reinforcement learning—to achieve state-of-the-art performance across UI drafting, polishing, and editing tasks.

**Compliance With Llm Reviewing Policy:**

Affirmed.

**Key Questions For Authors:**

please refer to the weaknesses.

**Limitations:**

yes

**Strengths And Weaknesses:**

Strengths：
+  The problem framing is highly practical. Shifting from one-shot generation to a closed-loop interactive visual optimization process directly addresses a fundamental mismatch in current UI-to-code models and successfully mirrors real-world, feedback-driven UI development workflows.
+ The formulation of Relative Visual Policy Optimization (RVPO) is a strong contribution. Deriving rewards from relative visual rankings via tournament-based aggregation elegantly bypasses the non-differentiability of the rendering space and mitigates the instability of absolute VLM scoring.
+ The research design incorporates a comprehensive training pipeline and introduces challenging new benchmarks, such as UI2Code-Real and UIPolish-bench, which ensure the model is tested against the complexity of in-the-wild webpages rather than just simplified synthetic data.
Weaknesses：
- This paper introduces Relative Visual Policy Optimization (RVPO) approximated via group-wise sampling and tournament aggregation, but computing pairwise comparisons for all ordered pairs within a sampled group creates an $O(N^2)$ computational bottleneck during the reinforcement learning phase. This heavy reliance on quadratic VLM inferencing severely limits the scalability of the training process for larger rollout sizes or larger datasets.
- This paper introduces rigorous visual evaluations leveraging both CLIP-based and VLM-based scoring to assess design fidelity and layout correctness, but completely omits developer-centric functional execution metrics. By relying exclusively on visual appearance, it remains unclear whether the generated DOM structures feature correct semantic HTML, responsive flexbox/grid behavior, and genuine interactivity, or if they are merely visually deceptive static mockups built with absolute positioning.
- This paper utilizes approximately 10M webpage image-HTML pairs collected via large-scale crawling from Common Crawl for continual pre-training, but lacks a rigorous and transparent data decontamination protocol regarding the downstream test sets (e.g., Design2Code, UI2Code-Real). This raises significant concerns about potential data leakage inflating the benchmark results.
- This paper presents UI Editing as one of the three core pillars of the interactive visual optimization paradigm , but the quantitative evaluation for this specific capability is surprisingly thin, relying entirely on a human evaluation of merely 69 tasks in UIEdit-Bench. This scale is insufficiently robust compared to the extensive automated testing provided for the drafting and polishing tasks.

---

> ### Author Rebuttal · Authors · 2026-03-31
>
> We sincerely appreciate your insights and are fully committed to addressing these points.
>
> ### **Response to W1: Scalability of Pairwise Comparisons in RVPO**
>
> We address the scalability concern with the following empirical evidence and implementation details:
>
> **(1) Pairwise Comparison is not the bottleneck**: The primary computational cost in our RL pipeline is the **autoregressive generation** and **rendering** of $N$ code candidates,  both of which are **parallelized** across worker nodes to minimize cost. The **pairwise VLM-based comparison** is a **post-hoc scoring step** performed only on successfully rendered outputs. Since each pairwise comparison is independent, this step is also **embarrassingly parallel**, meaning that the comparison overhead remains small, regardless of the total dataset size.
>
> **(1) Efficient Tournament-based Implementation**: In practice, a single **VLM inference** for the pair $(y_i, y_j)$ simultaneously determines both $o_{ij}$ and $o_{ji}$. Furthermore, as detailed in **Algorithm 1**, we only compare candidates that **pass the rendering check**. This significantly reduces the size of the comparison pool **$M$** ($M \le N$), making the actual number of inferences much smaller than the theoretical upper bound.
>
> **(3) Empirical Wall-clock Time**: To provide a concrete assessment, we measured the average wall-clock time per training iteration (with $N=16$, batch size 64). As shown in the following table, the pairwise comparison in RVPO accounts for only **2.2%** of the total time.
>
> | Stage      | Operation                           | Time (s) | Proportion (%) |
> |------------|-------------------------------------|----------|----------------|
> | Generation | Autoregressive token decoding       | 130.9s    | 75.4%          |
> | Rendering  | Headless browser execution          | 38.8s     | 22.4%          |
> | Comparison | Pairwise VLM evaluation (RVPO)      | 3.9s     | 2.2%           |
>
> ---
>
> ### **Response to W2: Addressing Functional Execution Concerns**
>
> We clarify that our generated outputs are not **static mockups** built with **absolute positioning**; instead, they are **functional**, **responsive HTML/CSS** code.
>
> **(1) Functional Integrity & Interactivity**: While our evaluation leverages CLIP/VLM-based scoring for automated benchmarking, the model is pre-trained and SFT-tuned on large-scale datasets containing complex interactivity and responsive layouts (Flexbox/Grid). As demonstrated in **https://jovial-sunburst-92285f.netlify.app**, the generated websites maintain **correct semantic HTML** and **genuine event-handling logic**. By incorporating interaction requirements with the target UI, the code generated by the model is fully capable of **functional interactivity**, enabling genuine UI interactions.
>
> **(2) Knowledge Retention in RL**:  During the RL phase, although we use visual rewards to fine-tune design fidelity, the foundational **coding logic** is preserved from the CPT and SFT stages. Our demo cases show that the model does not forget functional structure while optimizing for aesthetics.
>
> ---
> ### **Response to W3: Data decontamination and potential leakage**
>
> We would like to clarify that we have performed data decontamination to mitigate potential overlap between the crawl-based pre-training corpus, our SFT data, and downstream evaluation benchmarks. Specifically, we applied **image-level deduplication** using **image hashing** (e.g., perceptual hash-based filtering) to reduce near-duplicate samples. This approach ensures that visually similar images are identified and removed during data processing. Additionally, **all RL data** are disjoint from the evaluation benchmarks. We will revise the paper to include a more detailed explanation of the decontamination process to improve transparency.
>
> ---
>
> ### **Response to W4: UI Editing evaluation scale**
>
> We agree that the current **UI editing evaluation** is smaller in scale than **drafting/polishing** due to the difficulty of assessing **edit correctness**, **preservation of unedited regions** and **overall post-edit quality** with automatic metrics. We therefore rely on **human evaluation** for this setting.
>
> To strengthen the evidence, we expanded **UIEdit-Bench** from 69 to 100 tasks and re-ran human evaluation with the same three criteria: **edit correctness**, **preservation**, and **overall quality**. These results show consistent performance across the expanded evaluation set, and we will update the paper with the full evaluation on the 100-task benchmark in the revised version.
>
> | Model               | Edit Correctness (69 / 100) | Preservation (69 / 100) | Overall (69 / 100) |
> |---------------------|-----------------------------|-------------------------|---------------------|
> | Claude-4-Sonnet     | 4.83 / 4.79                | 4.54 / 4.44            | 4.69 / 4.62         |
> | UI2Code^N-RL         | 4.94 / 4.92                | 4.80 / 4.77            | 4.87 / 4.83         |

---

> > ### Author Rebuttal · Reviewer_ErBe · 2026-04-05
> >
> > I thank the rebuttal from authors, and I will maintain the score.

---

> > > ### Author Response · Authors · 2026-04-08
> > >
> > > Dear Reviewer,
> > >
> > > We sincerely appreciate your professional guidance, which has significantly improved the quality of our work. We are pleased that our additional ablation studies and clarifications regarding the efficiency of RVPO and structural integrity have fully addressed your concerns.
> > >
> > > We will ensure that all these clarifications, along with the expanded UIEdit-Bench, are incorporated into the final version of the manuscript. Thank you again for recognizing the value of UI2Code^N.
> > >
> > > Authors of UI2Code^N

---

### Decision · Program_Chairs · 2026-04-30

**Decision:**

Accept (regular)

**Comment:**

The paper proposes UI2Code^N, a framework that reformulates UI-to-code generation from a traditional single-pass task into an interactive visual optimization problem. To enable this feedback-driven iterative refinement, the authors introduce Relative Visual Policy Optimization (RVPO), a preference-based reinforcement learning method that optimizes relative visual rankings among rendered code candidates rather than relying on absolute scores.

Three reviewers unanimously praised the problem formulation, agreeing that shifting from single-pass generation to a closed-loop, interactive process successfully mirrors real-world frontend development workflows. The proposed RVPO method was highlighted as a clever, impactful, and strong contribution. Reviewers noted that RVPO elegantly bypasses the non-differentiability of the rendering space and mitigates the instability commonly associated with absolute VLM scoring. Additionally, the reviewers appreciated the comprehensive training pipeline, the well-written presentation, and the introduction of challenging new benchmarks like UI2Code-Real and UIPolish-bench.

However, the paper is not without weaknesses. The reviewers raised a number of issues. Here is a summary along the authors' rebuttals.
- Reviewer ErBe raised a concern that RVPO's pairwise comparisons would create a computational bottleneck during training. The authors provided empirical evidence showing that the pairwise VLM evaluation accounts for only 2.2% of the total wall-clock time per training iteration.
- Both Reviewer ErBe and Reviewer GpZw questioned whether relying exclusively on visual rewards might lead to "reward hacking," resulting in static mockups built heavily with absolute positioning rather than functional, responsive code. The authors quantitatively verified the structural integrity of their outputs, showing that the UI2Code^N-RL model uses a remarkably low absolute positioning ratio of 0.5%.
- Reviewer ErBe felt the UI Editing evaluation was too small at 69 tasks , and Reviewer GjW1 noted the comparison set lacked recent specialized systems. In response, the authors expanded the UIEdit-Bench to 100 tasks with consistent human evaluation results and provided new, favorable comparisons against baselines like UICopilot, DCGen, and ScreenCoder.
- Reviewer GpZw highlighted a severe transparency bottleneck due to a strict non-release policy for datasets and benchmarks. The authors successfully addressed this by committing to open-source the benchmarks, the trained models, the evaluation pipeline, and a heavily anonymized subset of the SFT data.

All three reviewers gave the paper a score of 4 (Weak Accept). Following the rebuttal, Reviewers ErBe, GjW1, and GpZw confirmed that the authors' additional experiments, clarifications, and open-source commitments resolved their respective concerns. Despite explicitly acknowledging that their issues were addressed, all reviewers chose to maintain their score of 4 for weak accept, which is an indication of a lack of champions for the paper, although the identified weaknesses were mostly neutralized during the discussion period.